



# Diabatic processes modulating the vertical structure of the jet stream above the cold front of an extratropical cyclone: sensitivity to deep convection schemes

Meryl Wimmer[1], Gwendal Rivière[2], Philippe Arbogast[3], Jean-Marcel Piriou[1], Julien Delanoë[4], Carole Labadie[1], Quitterie Cazenave[4], and Jacques Pelon[4]

[1]CNRM, Université de Toulouse, Météo-France, CNRS, Toulouse, France
[2]Laboratoire de Météorologie Dynamique/IPSL, Ecole Normale Supérieure, PSL Research University, Sorbonne University, École Polytechnique, IP Paris, CNRS, Paris, France
[3]Direction des Opérations pour la prévision, Météo-France, Toulouse, France
[4]LATMOS-IPSL, CNRS/INSU, University of Versailles, Guyancourt, France

**Correspondence:** Meryl Wimmer (meryl.wimmer@umr-cnrm.fr)

**Abstract.** The effect of deep convection parameterization on the jet stream above the cold front of an explosive extratropical cyclone is investigated in the global numerical weather prediction model ARPEGE, operational at Météo-France. Two hindcast simulations differing only in the deep convection scheme used are systematically compared with each other, with (re)-analysis datasets and with NAWDEX airborne observations.

The deep convection representation has an important effect on the vertical structure of the jet stream above the cold front at one-day lead time. The simulation with the less active scheme shows a deeper jet stream, associated with a stronger potential vorticity (PV) gradient in the jet core in middle troposphere. This is due to a larger deepening of the dynamical tropopause on the cold-air side of the jet and a higher PV destruction on the warm-air side, near 600 hPa. To better understand the origin of this stronger PV gradient, Lagrangian backward trajectories are computed.

On the cold-air side of the jet, numerous trajectories undergo a rapid ascent from the boundary layer to the mid levels in the simulation with the less active deep convection scheme, whereas they stay at mid levels in the other simulation. This ascent explains the higher PV noted on that side of the jet in the simulation with the less active deep convection scheme. These ascending air masses form mid-level ice clouds that are not observed in the microphysical retrievals from airborne radar-lidar measurements.

On the warm-air side of the jet, in the warm conveyor belt (WCB) ascending region, the Lagrangian trajectories with the less active deep convection scheme undergo a higher PV destruction due to a stronger heating occurring in the lower and middle troposphere. In contrast, in the simulation with the most active deep convection scheme, both the heating and PV destruction extend further up in the upper troposphere.





# 1 Introduction

Midlatitude high-impact weather (HIW) events are usually dynamically forced by near-tropopause disturbances and by specific configurations of the jet stream. Their surface imprints largely depend on the structure and intensity of the jet stream aloft. For instance, the rapid deepening of wind storms depends on the intensity of the jet stream (Wernli et al., 2002; Pinto et al., 2009; Rivière et al., 2010) and is favoured by the presence of a jet streak (Uccelini, 1990; Fink et al., 2009). As a second example, heavy precipitation and flood events are often forced by an elongated trough along the jet stream or a cut-off that just separated

from the jet stream following wave breaking (Massacand et al., 1998; Martius et al., 2008; Nuissier et al., 2011; Grams et al., 2014). Since the near-tropopause disturbance triggering the HIW event is often part of a Rossby wave train, the skills of NWP forecasts to accurately predict a HIW event depends on their ability to represent the troughs/ridges propagating along the jet stream during the days prior to the event (Parsons et al., 2017; Wirth et al., 2018). Consequently, there is a growing body of literature identifying systematic NWP biases along these downstream propagating near-tropopause wavelike disturbances

(Rodwell et al., 2013; Gray et al., 2014) and investigating dynamics of forecast errors along the midlatitude waveguide (Davies and Didone, 2013; Baumgart et al., 2018).

Looking at different NWP models, Gray et al. (2014) found systematic forecast errors in the jet representation increasing with forecast lead times. In particular, the potential vorticity (PV) gradient gets smoother on the poleward flank of ridges and Rossby wave amplitudes get smaller, the two being closely related (Harvey et al., 2016). Another consequence of the too smooth

PV gradient is the slowdown of phase speed with forecast lead time (Harvey et al., 2018). This is in that particular context, that the international field campaign NAWDEX (North Atlantic Waveguide Downstream and impact EXperiment) occurred in September-October 2016 (Schäfler et al., 2018). NAWDEX objective was to investigate the diabatic origin of forecast errors in the ascending part of extratropical cyclones along the so-called warm conveyor belts (WCBs), to analyse their downstream propagation along the waveguide and how they may affect the predictability of HIW events. Using NAWDEX observations

as a reference, Schäfler et al. (2020) showed underestimation of vertical wind shear in the vicinity of the tropopause in very short-term forecasts and analysed this could affect Rossby wave propagation.

Regions of strongest forecast errors and systematic analysis of forecast busts suggest that forecast errors originate from diabatic processes (Rodwell et al., 2013; Gray et al., 2014). Because of the PV invertibility properties and its conservation under adiabatic and frictionless processes, the PV perspective offers a classical and useful framework to investigate the influence of

diabatic processes on the atmospheric flow. The PV tracer technique that decomposes the PV rate of change into different model processes has been widely used during the last decade, mainly to study the near-tropopause PV anomalies associated with the jet stream (Chagnon et al., 2013; Martinez-Alvarado et al., 2014; Saffin et al., 2017; Spreitzer et al., 2019; Harvey et al., 2020) but also to study low-level PV anomalies associated with a surface cyclone (Crezee et al., 2017). The former cited studies found that near-tropopause PV is strongly affected by diabatic processes, mainly by latent heating, turbulence and

longwave radiation, and these processes maintain the strong PV gradient there. Saffin et al. (2017) showed that the decrease in tropopause sharpness with forecast lead time, originally diagnosed by Gray et al. (2014), is mainly due to the advection scheme and is only partially compensated by the increase in tropopause sharpness due to nonconservative processes.





The PV and potential temperature ($\theta$) Lagrangian framework can be used to explain atmospheric circulation differences between simulations performed with distinct models (Martinez-Alvarado et al., 2014) or between sensitivity numerical experiments made with the same model but using different parameterization schemes (Martinez-Alvarado and Plant, 2014; Joos and Forbes, 2016; Mazoyer et al., 2021; Rivière et al., 2021). Joos and Forbes (2016) used this approach to compare two simulations of the ECMWF-IFS global model with distinct cloud microphysics schemes. They found slight PV differences in the WCB outflow region amplifying on the downstream side of the ridge at 24 h-72 h lead time. Following a similar approach, Mazoyer et al. (2021) compared two simulations of a regional convection permitting model performed with two distinct cloud microphysics schemes. At 24 h-48 h lead time, the PV and wind speed differences were already well marked and anomalies span an extended band along the western edge of the ridge due to different heating rates within the WCB. The PV-$\theta$ framework has been also used to analyze WCB differences and impact on the tropopause with different deep convection parameterization schemes (Martinez-Alvarado and Plant, 2014; Rivière et al., 2021) but the amplitude of the impact on the upper-level circulation varies from case to case. Martinez-Alvarado and Plant (2014) found rather modest differences in the tropopause location after 24 h for a moderate cyclone between reduced and intense parameterized convection while Rivière et al. (2021) found important differences with a jet stream shift of a few hundred kilometers after 24 h for an explosive cyclone.

Recent NAWDEX-related studies have emphasized the importance of embedded convection within WCBs by comparing satellite observations and convective-permitting simulations to airborne radar measurements gathered during NAWDEX (Oertel et al., 2019, 2020, 2021; Blanchard et al., 2020, 2021). Oertel et al. (2020) and Blanchard et al. (2021) showed that the heating associated with embedded convection generates dipolar PV anomalies near the tropopause that reinforce the PV gradient and hence the jet stream. The ability of convectively created PV dipole to reinforce the jet depends on the region where convection occurs and on the vertical wind shear (Chagnon and Gray, 2009; Harvey et al., 2020; Oertel et al., 2021).

Following the same approach as in a companion paper (Rivière et al., 2021, hereafter RW21), the present study investigates the effect of parameterized deep convection on WCB and jet stream. RW21 compared three simulations of the Météo-France global model ARPEGE: two simulations were performed with two distinct deep convection parameterization schemes developed within ARPEGE, the one described in Bougeault (1985, thereafter B85) and the Prognostic Condensates Microphysics and Transport scheme of Piriou et al. (2007, thereafter PCMT). The third simulation was performed without any deep convection parameterization. Without deep convection parameterization the heating ahead of the cold front is organized in distinct cells of high values with a few degrees extent in longitude and latitude because convective instability is released at the resolved scales. In contrast, in presence of parameterized deep convection, the heating is much smoother because convective instability is released at subgrid scales. The consequence is that WCB ascents are more sustained with parameterized deep convection while they are more abrupt without. This regulating effect of deep convection parameterization on WCB was already noticed by Martinez-Alvarado and Plant (2014). However, this does not mean that the impact on the tropopause is stronger without deep convection parameterization. RW21 showed that the run in which deep convection is more active (B85) is also the one for which the heating extends further upward above the warm front of the extratropical cyclone and has a stronger PV destruction at upper levels. This leads to a distinct location of the jet stream compared to the other two runs, the one with PCMT deep convection scheme and the other without any active scheme. In RW21, the focus was on the WCB outflow region above the





bent-back warm front and the horizontal structure of the jet stream. In the present study, our aim is to analyse differences between the same simulations within the WCB ascending region above the cold front and to highlight differences in the vertical
structure of the jet stream.

The extratropical cyclone hereafter studied, called the "Stalactite cyclone" (1-4 October 2016), is of particular interest in a number of respects. It was formed off the Newfoundland coast and intensified over the North Atlantic with a deepening rate of 24 hPa in 24 h (Flack et al., 2021) as a classical bomb event (Sanders and Gyakum, 1980). It has been intensively observed during NAWDEX (Schäfler et al., 2018) by three flights: two flights of the French Falcon 20 from the Service des
Avions Français Instrumentés pour la Recherche en Environnement (SAFIRE) and one flight from the Deutsches Zentrum für Luft- und Raumfahrt (DLR) Dassault Falcon. Its development was accompanied by a burst in latent heating (Steinfeld et al., 2020) and the ridge building aloft led to the onset stage of the "Thor" block (Maddison et al., 2019) that last until the end of NAWDEX in mid-October 2016.

The paper is organized as follows. Section 2 presents the data and methods. It includes the description of the model simula-
tions and the main characteristics of the two deep convection schemes B85 and PCMT. It also provides information on various (re)-analysis datasets and on airborne observations made during the flight of the SAFIRE Falcon aircraft on 2 October over the ascending WCB region of the Stalactite cyclone. Finally, section 2 details the computation of PV-$\theta$ Lagrangian budgets. Section 3 shows differences in the jet stream representation between the different simulations and (re)-analysis datasets. Section 4 provides an explanation for these differences in terms of the PV-$\theta$ framework. Section 5 compares model simulations
to airborne observations to highlight the more realistic forecasts in the different regions. Finally section 6 is dedicated to the concluding remarks and discussion.

## 2  Data and method

### 2.1  Model and simulations set-up

As in the companion paper RW21, the study relies on simulations of the operational Météo-France global model, ARPEGE
(Action de Recherche Petite Echelle Grande Echelle; Courtier et al., 1991) and in particular on different members of its Ensemble Prediction System associated, called PEARP (Prévision d'Ensemble ARPEGE; Descamps et al., 2015).

For all PEARP members, the vertical resolution has 90 levels while the horizontal grid corresponding to T798 resolution, is stretched by a factor 2.4 and centred on France. Consequently, this resolution is about 10 km on France, 15 km on the zone of interest of this study and 60 km on the antipode of France. The time step of the model is 7.5 minutes. Models outputs are
available with a temporal resolution of 15 minutes and an horizontal resolution of 0.5°, while the vertical resolution is 50 hPa.

While operational PEARP members include perturbations in both model physics and initial conditions, the present study is based on PEARP reforecast dataset corresponding to ten members that have the same initial conditions (ARPEGE operational 4D-Var analysis at 12 UTC 1 October 2016) and only differ in their physics as in Ponzano et al. (2020), Binder et al. (2021) and RW21.





Two members amongst the ten ones are hereafter more deeply investigated. They only differ in their deep convection scheme: one with the Bougeault (1985) scheme (B85), the other with Prognostic Condensates Microphysics and Transport (PCMT; Piriou et al., 2007). They are referred as the REF and member 7 in Ponzano et al. (2020) and as B85 and PCMT in RW21. In addition, a third simulation, called NoConv, has no deep convection scheme activated. For more details on these three simulations, particularly concerning physical parameterization, reader is referred to RW21.

## 2.2 Differences between the two deep convection schemes

The two simulations studied in the present paper use two distinct parameterization schemes of deep convection, both being based on the mass-flux approach. Detrainment in the environment, precipitation and downdrafts phenomena are modelized in both B85 and PCMT. But in contrast to B85, the PCMT scheme is able to estimate the prognostic mixing ratios of the different hydrometeors inside the mass flux (updraft). It includes the same four hydrometeors (liquid cloud water, pristine ice, rain and

snow) involved in the large-scale cloud microphysical scheme of Lopez (2002), and the same microphysical phenomena, such as accretion, autoconversion, riming.

The two schemes are hereafter activated with two distinct closures. Bougeault (1985)'s scheme is activated with the convergence of total moisture fluxes (including both resolved and turbulent moisture fluxes) integrated from the surface to the considered level and when the atmosphere presents an unstable profile. In contrast, the Prognostic Condensates Microphysics

and Transport scheme (PCMT; Piriou et al., 2007) considers a CAPE closure.

## 2.3 Airborne observations and (re)-analyses

To better compare the effects of the deep convection scheme and to better estimate their realism, two types of references are used: observations from the NAWDEX IOP6, as well as operational analyses and reanalysis.

### 2.3.1 Observations from NAWDEX IOP6

The flight of the French Falcon 20 of SAFIRE, studied in the present study, occurred between 1301 UTC and 1616 UTC on 2 October during the NAWDEX field campaign (Schäfler et al., 2018). Figures 1a and b show the position of the flight according to the Stalactite cyclone. The aircraft took off at Keflavik, went south, realized a clockwise loop triangular in shape to the northeast of the Stalactite cyclone.

Figure 1b gives an insight of the cloudy region sampled by the flight. The major part of the flight occurred in the cloudy

region ahead of the cold front close to the cyclone center, which likely corresponds to the ascending part of the WCB. But the clear-sky zone appearing near the westernmost vertex of the triangle suggests that the flight crossed the cold front from east to west near 58° N.

During this flight, different measurements have been made. In-situ sensors on board the SAFIRE Falcon 20 measured pressure, wind and temperature at the flight level near 300 hPa. In addition, the RALI (RAdar-LIdar) platform developed

at LATMOS and DT-INSU, was on board the aircraft. This platform includes a multi-beam 95 GHz Doppler cloud radar



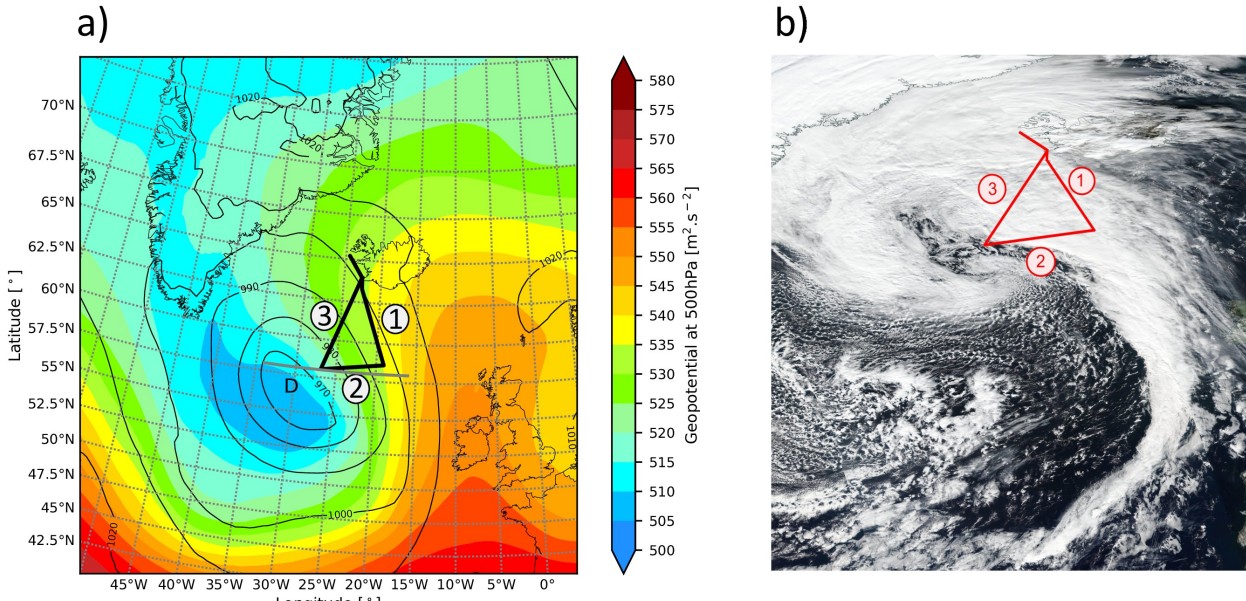

**Figure 1.** Visualisation of the Stalactite cyclone during its mature stage, the 2nd of October 2016: a) Geopotential at 500 hPa (shading), sea level pressure (black thin contour) at 12 UTC, Falcon flight (black bold line) and vertical cross section at 58° N (grey bold line) b) visible picture from VIIRS of the Suomi NPP satellite (NASA Worldview) with the Falcon flight in red.

(RASTA) (Radar Airborne System; Delanoe et al., 2013) and a Doppler high-spectral-resolution lidar (LNG; Bruneau et al., 2015). RASTA measures both reflectivity and Doppler velocity along three non colinear directions thanks to three downward antennas (nadir, backward and transverse). This configuration allows one to retrieve the 3D wind field in the vertical below the Falcon with a range resolution of 60 m and every 0.75 s leading to an horizontal resolution of about 300 m at the speed of the aircraft. The lidar operates at 532 nm and 1064 nm in backscatter mode only but measures Doppler velocity, polarisation and the backscattered light from molecules and particles separately at 355 nm. It gives information about optical parameters of aerosol and thin clouds together with along-sight wind below the aircraft at 15 m and 5 s range and time resolution respectively. Additional wind measurements were made by in-situ sensors at the aircraft altitude and dropsondes.

To better compare observations with model outputs, in-situ and RALI measurements have been averaged over intervals of 180 s to be close to the model grid spacing. Indeed, as the Falcon 20 has a mean speed of 200 m s$^{-1}$, it travels a distance of 36 km during 180 s, which it is close to the 0.5° horizontal grid spacing of the model outputs.

### 2.3.2 Operational analysis and reanalysis

Operational analyses of the ARPEGE and Integrated Forecasting System (IFS) models are considered. The same vertical resolution of 50 hPa and horizontal resolution of 0.5° than the ARPEGE simulations outputs are used. ERA5 reanalyses





 dataset (Hersbach et al., 2020) is also considered. However, as the original horizontal is about 0.25°, one grid point every two grid points is kept in order to get the same 0.5° resolution than the other simulations and analyses.

### 2.4 Lagrangian warm conveyor belt trajectories

#### 2.4.1 Initialization in the warm sector

The same forward trajectories as those shown in RW21 are used. They are initialized at 12 UTC on 1 October in the warm
sector of the extratropical cyclone and last 48 hours. To select WCB trajectories, a criterion of an ascent exceeding 300 hPa within 1 day during the period between 12 UTC on 1 October and 12 UTC on 3 October is applied.

#### 2.4.2 Initialization along the flight

To better characterize properties of the WCB air masses of the Stalactite Cyclone crossing the flight F7, another set of trajectories has been computed with the same trajectory algorithm as in RW21. It consists of 24 h backward and 24 h forward
trajectories starting from the flight legs over the whole vertical. For each flight leg, the trajectories are seeded on a vertical regular grid spacing of 12.5 hPa from 975 hPa to 200 hPa and a horizontal grid spacing of about 0.3° in longitude and latitude. As the flight lasted more than 3 h, trajectories from each flight leg must be seeded at a different time. The time of seeding is the time when the aircraft is in the middle of each leg. Hence, the first leg trajectories are seeded at 26.25 h forecast range, namely 1415 UTC on 2 October, the second leg trajectories at 27 h forecast range corresponding to 1500 UTC and finally the third leg
trajectories at 27.75 h forecast range, so 1545 UTC.

Overall, 5292 trajectories lasting 48 h have been computed with 5292 seeding points along the flight path (63 in the vertical × 84 in the horizontal). To prevent trajectories to cross the surface, pressure is limited at 975 hPa. Trajectories with a minimum ascending rate of 300 hPa in 24 h are considered as belonging to the WCB. It leads to 1870 WCB trajectories for B85 and 1972 WCB trajectories for PCMT.

### 2.5 Heating and PV tendencies

As in RW21, the heating $\dot{\theta}$ is computed into two different manners. The first method uses instantaneous temperature tendency datasets associated with the different diabatic processes parameterized in the model (large-scale cloud microphysics, convection, radiation, turbulence). These temperature tendencies are first provided on the ARPEGE stretched grid and model levels before being interpolated on the 0.5° × 0.5° horizontal grid and pressure levels. The second method computes the heating
using centred finite-difference schemes applied to the potential temperature at the resolution of the model outputs chosen for this study: 0.5° × 0.5° in the horizontal, 50 hPa in the vertical and 15 minutes in time. Since the Lagrangian trajectories are also computed on the latter grid, the variations in $\theta$ along trajectories is very close to the integrated heating $\dot{\theta}$ obtained with the second method (not shown). The first method only roughly approximates the variations in $\theta$ along trajectories for mainly two reasons: firstly, the dynamical core of ARPEGE does not strictly conserve $\theta$ because of numerical diffusion in the advection
scheme and secondly the various interpolation steps in the offline trajectory algorithm to get the temperature tendency terms





on the model output grid generate uncertainties. Both methods are hereafter used and information on the choice of the method is provided in the captions of the figures: the second method has the advantage to nearly close the heating budget while the first method has the advantage to provide a decomposition of the heating into various diabatic processes.

As the PV tendency depends on spatial variations of the heating and frictional terms, their computation is made by applying finite-difference schemes to the heating and frictional terms. The frictional terms in the zonal and meridional momentum equations are made available in the stretched/rotated Gaussian reduced model grid and at model levels. They are first interpolated on the $0.5° \times 0.5°$ horizontal and 50 hPa vertical grids before applying the finite-difference schemes to them. The heating is computed following the second method described above as it leads to a much more accurate approximation of the total PV tendency than the first method.

## 3 Impact of deep convection representation on the Stalactite Cyclone dynamic

The vertical structure of the jet stream at 58° N is shown for the three simulations and different (re)-analysis datasets in Fig. 2. This latitude roughly corresponds to the southern leg of the flight (see grey line in Fig. 1a) and to the western edge of the upper-level ridge. In the three references (i.e. the two analyses and ERA5), the maximum wind speed is located near 24° W between 300 hPa and 400 hPa at the interface between stratospheric and tropospheric air and varies between 55 m s$^{-1}$ and 60 m s$^{-1}$. The height of maximum wind speed fluctuates from case to case: it is about 340 hPa in the two analyses while it is near 390 hPa in ERA5. Such differences in wind speed are accompanied by similar differences in PV: the dynamic tropopause (2 PVU isoline) descends until about 400 hPa in the two analyses while it descends further down to 500 hPa in ERA5. On the tropospheric side, east of the jet maximum, stronger negative values (more than $-0.5$ PVU at 23° W) appear in ERA5 than in the two analyses. The jet is narrower and slightly deeper in ERA5 than in the analyses, consistent with stronger PV gradient between 400 hPa and 500 hPa in the former than in the latter datasets.

The three ARPEGE forecasts with distinct deep convection representation simulate the speed and position of the jet stream reasonably well in comparison with the three references: the jet is centred at 24° W with also a maximum between 50 and 60 m s$^{-1}$ in both simulations. However, the vertical structure differs from one run to another. PCMT and NoConv simulate a deeper jet stream with a center located at 390 hPa with wind speed values up to 40 m s$^{-1}$ reaching 650 hPa. In contrast, in B85, such high wind speed values do not go further down than 575 hPa. The deeper jet stream in PCMT and NoConv is associated with a lower tropopause to the west, as it goes down to 650 hPa in both PCMT and NoConv and only to 400 hPa in B85. It is also associated with negative PV values going further down to the east: the area of negative PV values extends from 350 hPa to 525 hPa in PCMT and from 350 hPa to 475 hPa in NoConv while it goes from 450 hPa to even higher than 250 hPa in B85.

As the largest differences between the two forecasts appear in the middle of the troposphere, horizontal cross sections of wind speed and potential vorticity are shown at 600 hPa in Fig. 3. In all simulations (references as well as forecasts), wind speed values higher than 40 m s$^{-1}$, corresponding to the lower part of the jet stream, are located above and along the cold front of the Stalactite cyclone which is noticeable by the change in curvature of the sea level pressure contours along an axis oriented from southeast to northwest.

**Figure 2.** Vertical cross section at 58° N (grey line in Fig. 1 of the zonal wind (shadings) and Potential Vorticity (black contours with hatched areas for values superior to 2PVU) at 15 UTC, 2 October 2016 for a) IFS analysis, b) ARPEGE analysis, c) ERA reanalysis, d) simulation with B85, e) simulation with PCMT and f) simulation without deep convection parameterization. The thick crosses represent the location of wind speed maxima.

Among the six datasets, PCMT and Noconv are the two simulations exhibiting the most intense jet with wind speed values beyond 40 m s$^{-1}$ from 18° W to 26° W and between 55° N and 58° N, located east of a band of PV values exceeding 2 PVU. The other datasets do not exhibit PV values as large as in PCMT and NoConv at 600 hPa in that region close to the westernmost vertex of the triangular-shaped flight. NoConv is the run showing the highest PV values to the west of that primary jet. More to the east, near the easternmost vertex, a less well-defined secondary jet with values close to 30 m s$^{-1}$ appears in some datasets.



**Table 1.** Root mean square of the difference with ERA5 reanalysis of the IFS and ARPEGE analyses and the three forecasts in PV and wind speed at 600 hPa over the domain shown in Fig. 3.

|  | ECMWF-IFS analysis | ARPEGE analysis | B85 | PCMT | NoConv |
|---|---|---|---|---|---|
| wind speed (m.s$^{-1}$) | 1.85 | 2.37 | 2.95 | 2.99 | 3.20 |
| PV (PVU) | 0.31 | 0.33 | 0.34 | 0.38 | 0.47 |

It has different shape and extension in the different datasets. For instance, this secondary jet in B85 extends further to the

northwest than in PCMT or in ECMWF-IFS and ERA5. The ARPEGE analysis brings similarities with B85, which is not surprising as they both use the same model and the same deep convection scheme (B85). ERA5 and ECMWF-IFS analysis are similar too as they both use the IFS model.

This is confirmed by computing root-mean-square (RMS) of the differences of each dataset with ERA5: the lowest RMS is with ECMWF-IFS, the second lower is ARPEGE analysis, then B85, PCMT and NoConv in ascending order (table 1).

Since the two analyses have lower RMS difference with respect to ERA5 than the forecasts, it gives confidence in assessing the performance of the three forecasts as the three references are closer to each other than to the forecasts. Among the three forecasts, NoConv is the one leading to the highest RMS and in that sense it is less skillful than the other two. PCMT and B85 have rather similar RMS values with those of PCMT being slightly higher. In terms of PV, it is clearly the band of high PV values to the west of the primary jet that increases the RMS error of PCMT with respect to ERA5, and more importantly that of

NoConv. But PCMT does not perform so differently from B85 because B85 has other significant differences with ERA5 along the third leg of the flight and related to the too far northwestward extended secondary jet as confirmed in section 5.

The other PEARP members, differing only in their physical parameterization of deep convection, turbulence, shallow convection and surface oceanic fluxes, as described in RW21, are also compared in an additional sensitivity study (Figs. S1 and S2). Members 1, 2, 4, 5, and 9 predict a maximum of the jet stream near 300 hPa as the B85 simulation (member 0) (Fig. S1).

In contrast, members 3, 6, 8 are marked by a jet maximum located, lower, between 350 hPa and 400 hPa and, in that sense, behave as in the PCMT simulation (member 7). Furthermore, the first group of members does not show any stratospheric air with a PV superior to 2 PVU in middle troposphere (600 hPa) whereas the second group shows systematic areas with PV higher than 2 PVU (Fig. S2).

Notice that the common point of each group is the type of closure used in the deep convection schemes. Members 0, 1,

2, 4, 5 and 9 share the same deep convection scheme, namely B85 with a moisture convergence closure. In the other group, all members share a CAPE closure with members 6, 7 and 8 using the PCMT scheme and member 3 using the B85 scheme. By analyzing hindcasts of heavy precipitation events, Ponzano et al. (2020) also emphasized a clustering of the 10 PEARP members into 2 groups but their partition was dependent on the deep convection used (B85 vs PCMT). In the present case as well as in RW21, the separation more clearly emerges according to the convection-parameterization closure (moisture vs

CAPE).



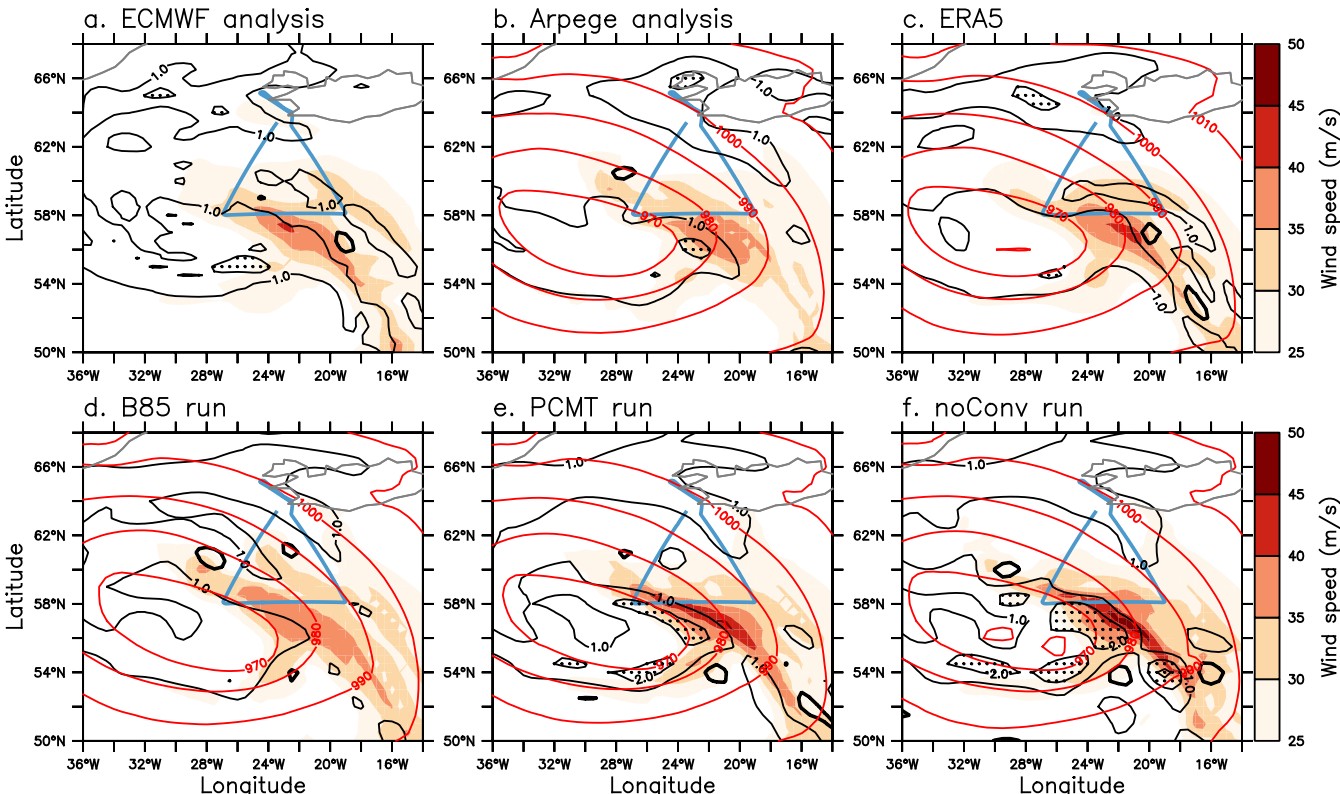

**Figure 3.** Wind (shadings) and Potential Vorticity (black contours with hatched areas for values superior to 2 PVU) at 600 hPa with sea level pressure (red contours) at 15 UTC, 2 October 2016 for a) IFS analysis, b) ARPEGE analysis, c) ERA reanalysis, d) simulation with B85, e) simulation with PCMT and f) simulation without any deep convection representation. The Flight F7 of the SAFIRE Falcon appears in blue line.

To conclude, PCMT has a deeper jet stream than B85 over the cold front in association with higher positive PV values to the west and smaller negative PV values to the east in the mid troposphere around 600 hPa. At this stage, it is rather difficult to determine which deep convection scheme is more realistic because the height and vertical structure of the jet in the three references (two analyses and ERA5) are usually in between the two runs. Since the two simulations with activated deep convection scheme behave in opposite ways and exhibit a large difference in the vertical structure of the jet stream, it is worth investigating the reasons of this difference as done in next sections, in particular by analysing the properties of the WCB.

## 4   Heating and Potential Vorticity differences in the Warm Conveyor Belt

Since the main difference in the jet stream highlighted in the previous section occurs during and near the SAFIRE flight on 2 October afternoon, the air streams crossing the flight are studied in the present section. They are modelized by 48 h Lagrangian trajectories centred on the flight (see subsection 2.4.2 for their definition). Trajectories satisfying the WCB criterion (300 hPa





ascent in 24 h) are represented in Fig. 4a for B85 and Fig. 4b for PCMT. All WCB trajectories have a poleward direction along the cold front and then may turn cyclonically or anticyclonically as in the classical picture of the WCB (Schemm et al., 2013; Martinez-Alvarado et al., 2014). Those having a cyclonic curvature move towards the cyclone center, while the majority of them have a pronounced anticyclonic curvature and orient towards Scandinavia at the end. However, there is no separation between the cyclonic and anticyclonic branches of the WCB but rather a continuum exists between cyclonically- and anticyclonically-curved trajectories. Also, many trajectories first curve cyclonically before curving anticyclonically.

More cyclonic trajectories are present near the minimum of pressure in PCMT than B85. Computation of the curvature during the last 3 h of the trajectories confirms it: 76% of them have an anticyclonic curvature during that last 3 h and so 24% are cyclonic in B85 while the proportions are quite different in PCMT with 72% of anticyclonic trajectories and 28% of cyclonic ones. Finally, some slight differences are noticed in the anticyclonic branch with more trajectories moving further south of Scandinavia in PCMT than in B85.

The pressure along these trajectories represented in color, shows that trajectories ascend as they move northward. For both simulations, some ascending regions are located far south of the flight (e.g., near 25° W; 50° N), some others further north (40° W; 62.5° N) but most of them are in the vicinity of the flight. Hence, the flight clearly occurs in the main ascending region of the WCB.

Latitude-pressure plots of the trajectories colored by the heating rate $\dot{\theta}$ are shown in Fig. 4c for B85 and Fig. 4d for PCMT. In both simulations, the maximum heating undergone by the trajectories is about 2 K h$^{-1}$ and logically occurs in the ascending part of the trajectories. Some cooling stage occurs in the lower troposphere before the trajectories reach the freezing point 0° C (purple dots) due to evaporative/melting processes. In B85, strong cooling is obvious just below the freezing point between 45° N and 50° N that is likely due to snow melting. Another slight cooling area appears in both simulations: in the upper troposphere due to long wave radiation. Some large differences also exist between the two simulations. A large part of the PCMT trajectories present a strong heating of 2 K h$^{-1}$ in the positive temperature area, while this phenomenon is much more reduced in B85, questioning the different behaviour between these two convection schemes in the liquid phase. This more intense heating occurring sooner along the trajectories and at lower altitude in PCMT has some implications in terms of PV tendencies as shown later.

Figures 5a and 5b show the wind speed at 15 UTC on 2 October 2016 along the last half of the flight for B85 and PCMT respectively, together with the difference in PV between PCMT and B85 (PCMT-B85) in each panel. Additionally, the positions of the WCB trajectories initialized along the legs of the flight are represented by the grey circles. Only the second and third leg of the flight F7 are considered as very few trajectories satisfying the WCB criterion cross the first leg. Note that the abscissa is not the time but a trajectory index, which is the number of trajectory seeds, on the horizontal, along the flight. Between 300 hPa and 500 hPa, dipolar PV anomalies appear in the vicinity of the wind speed maxima with positive values to the east and anticyclonic to the west. It means that the PV gradient is stronger in B85 in the upper troposphere and is logically associated with stronger wind speed maxima at those levels. Between 500 hPa and 700 hPa, opposite-sign PV anomalies also appear on both sides of the jet, but here the positive values are to the west and negative ones to the east of the jet. It means there is a stronger PV gradient, which is associated with a stronger wind speed maxima in the mid troposphere and thus a

**Figure 4.** Upper panels: longitude-latitude plots of the pressure (shadings) along the warm conveyor belt trajectories crossing the flight F7 for a) B85 and b) PCMT. Lower panels: vertical point of view (Latitude-Pressure plots) heating (shadings; 1st method of computation) along the warm conveyor belt trajectories crossing F7 for c) B85 and d) PCMT. Intersection with the iso-0° C is represented by the purple dots.

deeper jet for PCMT. At the same levels but further away from the jet, the PV anomaly changes sign again (see trajectory index higher than 70). The opposite-sign PV anomalies centred at trajectory index 70 reinforce the PV gradient in B85 with respect to PCMT and leads to the presence of a secondary jet at those levels for the former run, which has been already discussed when commenting Fig. 3d. Therefore, between 500 hPa and 700 hPa, the PV difference exhibits a tripolar anomaly in each leg, which is systematic of all sections crossing the cold front from 12° W-50° N to 28° W-62° N (Figs.5c-d).





The positions of the WCB trajectories are located between 900 hPa and 300 hPa for both simulations, but they are more numerous in the upper layer between 400 hPa and 300 hPa in B85, particularly in leg 2 (compare Figs. 5a and b). The more numerous upper-level WCB trajectories in B85 are in a positive PV difference, which means a lower PV in B85. As a strong heating occurs between 800 hPa and 400 hPa followed by a rapid decrease above 400 hPa (see Figs. 4c and d), it indicates

that the vertical gradient of the heating is negative at 400 hPa and above. Hence, the WCB trajectories at the time of the flight undergo a negative PV tendency above 400 hPa and the more numerous WCB trajectories in B85 at those levels induce more PV destruction than those in PCMT. It provides an explanation for the smaller PV east of the jet, stronger PV gradient, and stronger wind speed in B85 at pressure lower than 400 hPa. Between 500 hPa and 700 hPa along the cold front, most of the WCB trajectories initialized in the warm sector are located in the negative PV anomaly (Figs. 5c,d). Very few of them,

initialized along the flight, are within the positive PV anomaly to the east of the main jet (see trajectory index between 45 and 60 in Figs. 5a-b).

To better explain the deeper jet stream in PCMT, the next section analyses the origin of the positive PV anomaly (black dots in Figs. 5a-b) and negative PV anomaly (green dots) located on both sides of the jet stream.

### 4.1 Positive PV anomaly on the cold-air side of the jet

To investigate the origin of the positive PV anomaly, the 24 h backward trajectories, whose seeding point is represented by black dots on Figs. 5a and b, are considered. Figure 6a shows the time evolution of PV averaged over all trajectories reaching the positive PV anomaly along legs 2 and 3 (red and orange respectively for B85; blue and cyan respectively for PCMT). The first striking result is that the averaged PV is almost the same at the initial time (16 UTC on 1 October) between the two simulations while they differ by about 0.3 PVU at the final time (16 UTC on 2 October). It clearly shows that the higher PV

in PCMT than B85 at the time of the flight (1500 UTC and 1545 UTC on 2 October for leg 2 and 3 resp.) is solely due to diabatic PV modification along trajectories. More precisely it is between 00 UTC and 04 UTC on 2 October that the B85 and PCMT curves move away from each other (compare the red and blue curves or the orange and cyan ones). After 04 UTC, the PV difference B85 and PCMT is maintained whatever the leg, even though the separation distance between the curves may undergo some fluctuations, as at 11 UTC for leg 2.

The time evolution of the PV tendencies computed by summing the tendencies due to heating and friction is shown in Fig. 6b for leg2 (bold solid lines). The good correspondence between the sign of that sum (Fig. 6b) and the slope of the PV evolution (Fig. 6a) shows that the budget is correctly done. For B85, between 16 UTC and 22 UTC, PV increases and the sum of all terms is positive while after 22 UTC, PV tends to slightly decrease consistent with near zero or negative PV tendency. Only at later times, after 12 UTC on 2 October, PV increases a little bit. For PCMT, the tendency is first near zero and the PV does not

change much prior to 20 UTC on 1 October. But after that short period, the PV tendency is quasi-systematically positive and higher than that of B85 except near 03 UTC on 2 October or 10 UTC on 2 October.

The PV tendencies decomposition into heating (thin line) and friction (thin dashed line) parts clearly shows that the PV fluctuations are dominated by the heating. In particular, positive PV tendencies are solely due to the heating. Figures 6c and d represent the vertical profiles of PV tendencies and heating averaged over all grid points where there is a WCB trajectory from

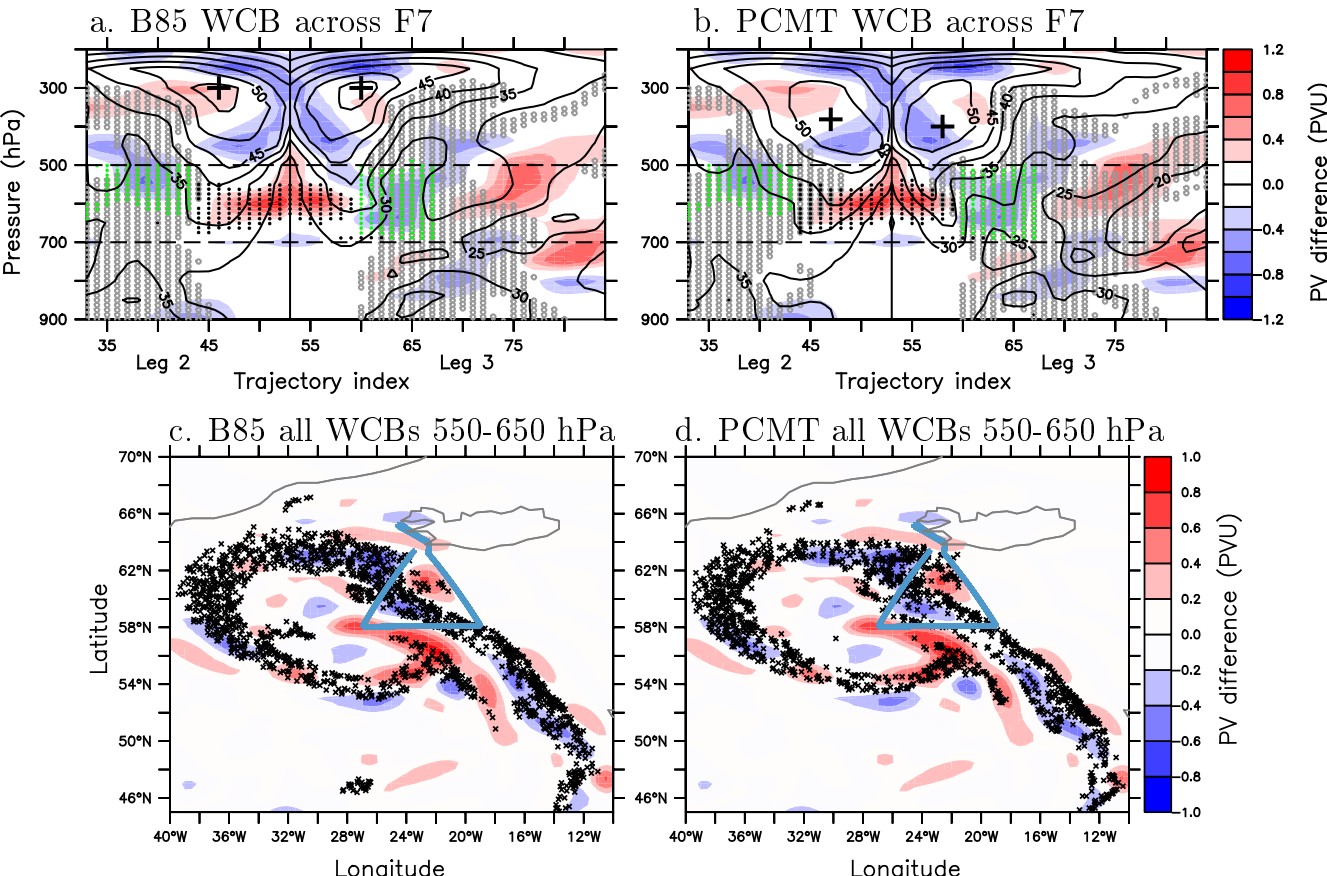

**Figure 5.** Upper panels: Vertical cross section of the difference (PCMT-B85) in PV (shadings) along the second and third leg of flight F7 at 15 UTC, 2 October 2016. The wind speed (black contours) and intersection of WCB trajectories with an ascending of 300 hPa in 24 h with F7 (grey circles) are shown for a) B85 and b) PCMT. In panels (a) and (b), the thick crosses represent the wind speed maxima in legs 2 and 3 for B85 and PCMT respectively. Trajectories where PV anomalies are positive and negative between 500 hPa and 700 hPa are in black and green dots respectively. Lower panels: PV difference at 600 hPa (shadings) and WCB trajectories, initialized in the warm sector, positions between 550-650 hPa (crosses) at 15 UTC 2 October 2016 for c) B85 and d) PCMT. The flight is in blue line.

leg2 for B85 and PCMT respectively. One large difference between the two figures concerns the pressure distribution along the trajectories (red or blue dashed curves). For B85, they are clustered in a single group always transported in the middle of the troposphere during 24 hours before reaching the positive anomaly of leg 2 (see red dashed curves in Fig. 6c). Depending on how they are positioned relative to the heating/cooling regions, they may undergo PV increase or decrease. For instance, the slight PV increase between 16 UTC and 22 UTC on 1 October or after 12 UTC on 2 October is explained by the fact that

the trajectories are above a cooling region. In contrast, for PCMT, the trajectories are clustered into two well separated groups, one half transported in the middle of the troposphere as in B85 but the other half starting in the boundary layer and suddenly rising at 04 UTC 2 October (see blue dashed curves in Fig. 6d). Slightly before and during the beginning of the ascents of the

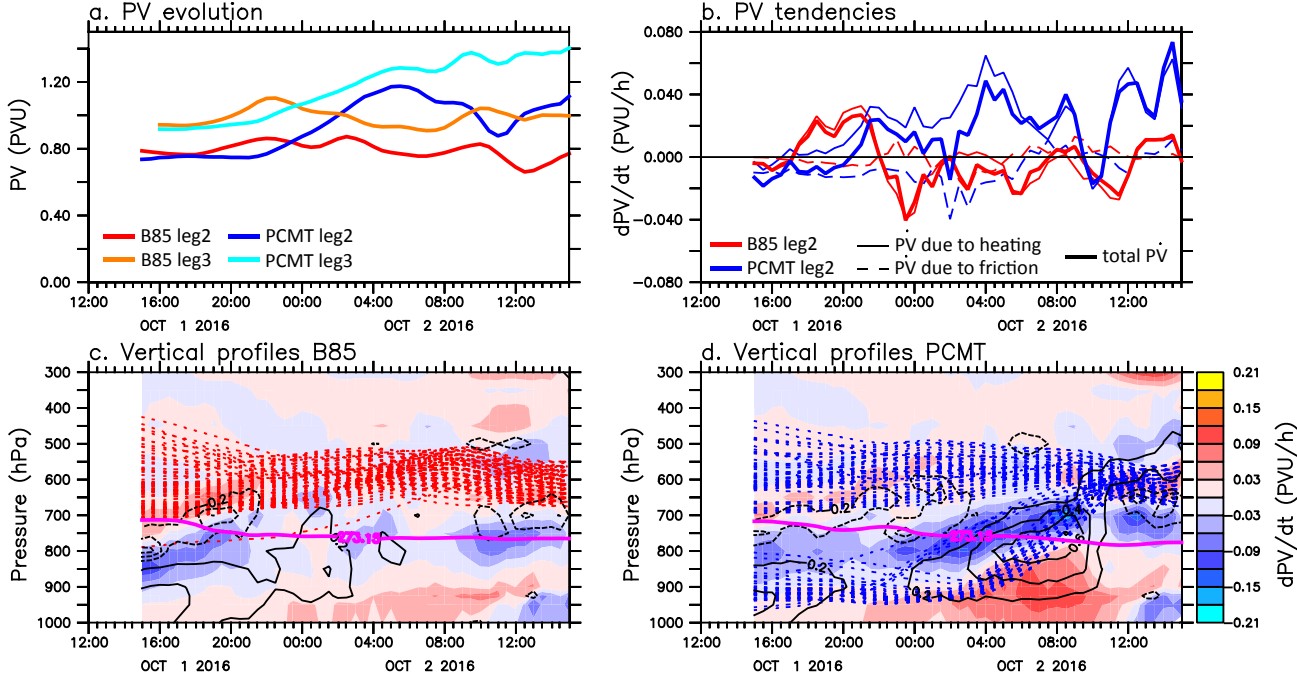

**Figure 6.** a) Time evolution of PV for WCB trajectories reaching the positive PV anomaly in leg 2 (red for B85 and blue for PCMT) and leg 3 (orange for B85 and light blue for PCMT); b) Time evolution of PV tendencies due to heating (thin solid lines), friction (thin dashed lines) and total (bold solid lines) for B85 (red) and PCMT (blue); Vertical profiles, according to time, of heating (contours, dashed and solid for negative and positive values resp.; second method of computation) and PV tendency due to heating (shadings) along WCB trajectories reaching the positive PV anomaly for c) B85 (red dashed curves) and d) PCMT (blue dashed curves) respectively. The iso-0° C is the purple line.

latter trajectories (i.e. from 00 UTC to 04 UTC on 2 October), PV rapidly increases because the trajectories are below a region of strong heating. This is precisely during this period, that the two averaged PV in B85 and PCMT get away from each other

(Fig. 6a) and the PV tendencies are largely different (Fig. 6b). When the same trajectories go above the heating, they undergo a short period of PV decrease between 08 UTC and 10 UTC but they rapidly go above a cooling region after 12 UTC when they catch up the first group of trajectories and their PV increases once again.

Figure 7 helps to further visualize the position of the trajectories with respect to the heating when the two PV start to get away from each other (Fig. 6a). For that purpose, the chosen time is 03 UTC on 2 October. As previously observed in Fig. 6c,

WCB trajectories from the positive anomaly do not present any ascent in B85 (Fig. 7a). They are all transported west of the heating area behind the cold front around 650 hPa and the iso-304 K (Figs. 7a,c). In PCMT, the group of trajectories around 650 hPa have more or less the same position relative to the main heating area as the one in B85 (Figs. 7b,d). However they are above a well marked cooling region near 24-26° W and 700 hPa and undergo a slightly larger PV increase than those for B85.





**Figure 7.** Pressure (shadings) along WCB trajectories reaching the positive PV anomaly in leg 2 for a) B85 and b) PCMT respectively, heating (black contours ; units : 0.4 K h$^1$) and WCB position (blue crosses) at 03 UTC, 2 October; Vertical cross sections of heating averaged between 45° N and 49° N (black contours; second method of computation), potential temperature (red contours) and WCB positions (blue crosses) at 03 UTC 2 October, for c) B85 and d) PCMT.

The heating budgets in Figs. S3 and S4 for B85 and PCMT respectively show that the cooling region is due to both radiation
and turbulence at the top of mid-level convective clouds in the cold sector of the cyclone. This increase in PV for trajectories moving in the mid troposphere over the cold sector can be seen at different time intervals in both runs (between 15 UTC on 1 October and 00 UTC on 2 October or after 12 UTC on 2 October in Figs. 6c,d) but the PV increase appears to be more important in average in PCMT (compare the reddish colors between 700 hPa and 600 hPa in Figs. 6c and d). The other group of trajectories found in PCMT is located at 22° W and 900 hPa. They are located in the lowest and most western part of the
main heating area ahead of the cold front, which is dominated by large-scale cloud heating (Fig. S4), and will rapidly ascent during the following hours (Fig. 6d).





To conclude, the higher PV obtained in PCMT than B85 on the cold-air side of the jet stream is mainly due to diabatic processes occurring between 00 UTC and 04 UTC 2 October during which half of the PCMT trajectories rapidly ascend and undergo a PV increase below a strong heating area. This heating is mainly due to large-scale latent heating and, to a lesser

extent, due to convection heating (Figs. S3, S4). Some of these trajectories exceeding 300 hPa ascent in 24 h satisfy the WCB criterion and are thus identified with grey circles in Fig. 5b near trajectory indexes 45-50. An additional factor to explain the difference in PV between the two runs concerns the group of trajectories evolving in the middle of the troposphere: in PCMT, they are more often subject to PV increase in presence of cooling areas below them at the top of convective mid-level clouds in the cold sector of the cyclone. In B85, this happens less regularly along similar trajectories.

## 4.2  Negative PV anomaly on the warm-air side of the jet

The same approach based on backward trajectory is adopted to better understand the origin of the negative PV anomaly to the east of the jet, which is mainly embedded in the WCB region (Figs. 5a-b). At the time of the flight (1500 UTC and 1545 UTC on 2 October for leg 2 and 3 resp.), the averaged PV is about 0.3 PVU lower in PCMT than in B85 whatever the leg (Fig. 8a). For leg 3, the difference in PV rapidly increases from 04 UTC to 08 UTC on 2 October while for leg 2, it increases from 12

UTC to 16 UTC on 2 October. Even though the timing is different, for both legs the PV difference is small at the initial time (16 UTC on 1 October) and the PV difference has a diabatic origin occurring during the last 12 hours before reaching the flight legs.

Let us now focus on the negative PV anomaly of leg 2. For both PCMT and B85, PV first increases and then decreases. However, the PV variations are larger in PCMT than B85. This difference is due to the heating contribution in the PV tendency

budget and not due to the friction contribution (Fig. 8b). The heating being stronger in PCMT than B85 (maxima are about 2 K h$^{-1}$ for PCMT and 1.4 K h$^{-1}$ for B85), its gradient is stronger leading to higher amplitude PV tendency in the former case (Figs. 8c-d). The heating is also more vertically stacked in the lower troposphere in PCMT in such a way that the trajectories are already all advected in a region above the heating maximum in PCMT after 10 UTC 2 October and undergo a negative PV tendency (Figs. 8b,d). During this later period, the B85 trajectories are advected near a region of maximum heating and have

thus near zero PV tendency (Figs. 8b,c).

Figure 9 provides a 3D picture of the position of the trajectories with respect to the heating at 12 UTC on 2 October, i.e. the time when the PV difference between PCMT and B85 starts to increase. At the initial time, a majority of trajectories were located in the boundary layer of the warm sector of the cyclone whatever the simulation (Figs. 9a-b). At the time of the plot, all trajectories lie within the strong heating region ahead of the cold front. However, their positions relative to the heating

vertical gradient largely differ between PCMT and B85. This is mainly due to two distinct features in the heating fields. The B85 heating extends more to the upper troposphere and is more vertical whereas PCMT heating is more confined in the middle troposphere and is marked by an eastward tilt with height. The latter tilt is rather systematic whatever the time chosen (see Figs. 7d and 9d). These two distinct features place the trajectories in a region of negative heating gradient and thus negative PV tendency in PCMT while the heating gradient is weak for the B85 trajectories. Thus, contrarily to B85, the PCMT trajectories

are already passing over the main heating area ahead of the cold front and already loose PV before reaching the flight leg.



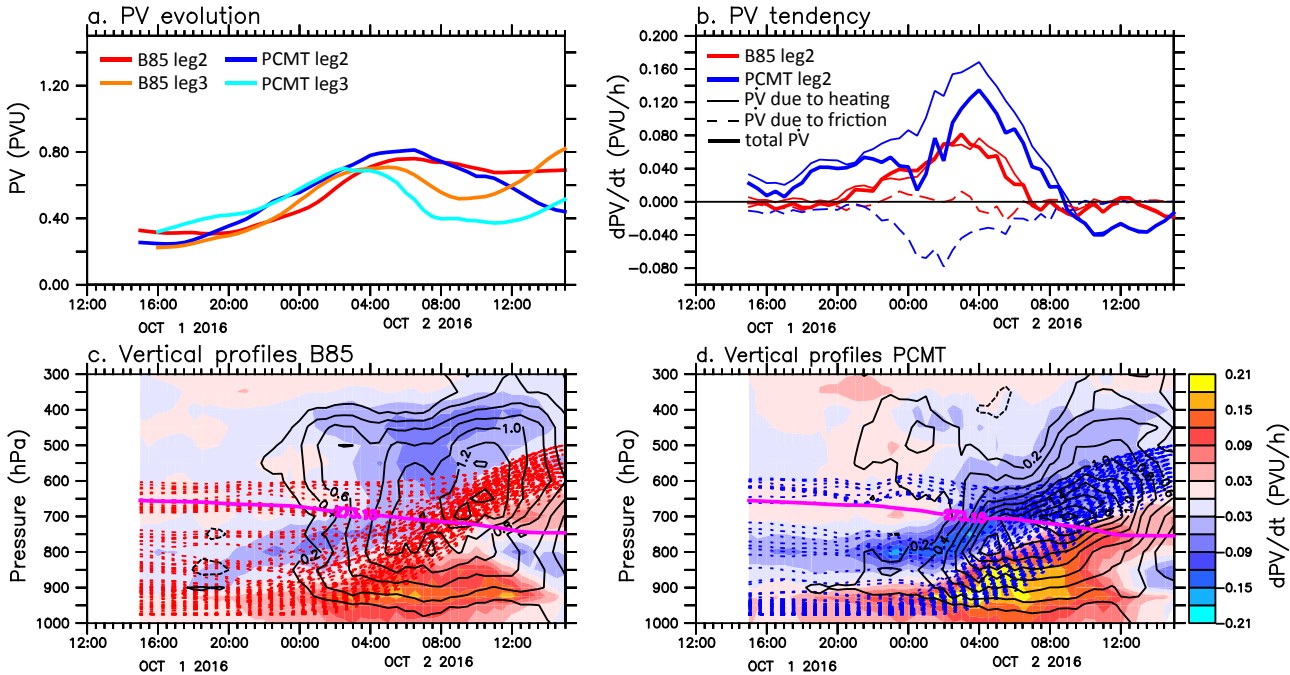

**Figure 8.** Same as Figure 6 but for the trajectories reaching the negative PV anomaly shown in Figures 5a,b.

To summarize, the deeper jet stream in PCMT than B85 can be explained by distinct diabatic processes occuring on both sides of the jet in the middle troposphere. On the cold-air side, half of the PCMT trajectories undergo some PV increase as they travel below the heating before reaching the middle troposphere while all B85 trajectories keep travelling in the middle troposphere. On the warm-air side, PCMT trajectories undergo a more rapid PV decrease because they already pass over the heating which is more confined at lower levels. This induces a PV difference between PCMT and B85 that reinforces the PV gradient near the jet core and thus the jet in PCMT relative to B85.

## 5 Comparison with observations from the NAWDEX IOP6

Since it is the difference in the heating structure that makes the difference in the vertical structure of the jet stream, and since the heating is linked to cloud formation and microphysics, a comparison is made between the ice water content (IWC) of the model simulations and the one retrieved from the radar-lidar observations in Fig. 10.

Only the second half of the flight is considered. To better compare to ARPEGE simulations, observations are interpolated at the model outputs resolution (0.5° grid spacing roughly corresponding to 180 s at the aircraft speed) in Fig. 10. Two IWC products are retrieved from the observations using the Varcloud algorithm (Delanoë and Hogan, 2008; Cazenave et al., 2019): one is based on the radar RASTA measurements only (Fig. 10a) ingesting both reflectivity and Doppler velocity, and the other





**Figure 9.** Same as Fig. 7 but for the trajectories reaching the negative PV anomaly in leg 2 shown in Figs. 5a,b and at 12 UTC 2 October.

on RALI (radar and lidar) measurements, i.e. assimilating radar reflectivity and lidar backscatter (Fig. 10b). Since the lidar is more sensitive to small particles and small hydrometeors contents, the RALI retrieval usually leads to smaller IWC than the RASTA retrieval as confirmed by comparing Figs. 10a and b. Note that these two retrievals do not use the same microphysical assumptions due to the difference in sensitivity and penetration capability of these two instruments. As described in Cazenave et al. (2019), the main uncertainties in the retrieval come from the mass-size, area-size relationships. Therefore, the comparison

between these two retrievals gives an idea of the uncertainties related to those retrievals.

As the flight crosses the WCB region twice, two zones with high IWC are observed in each retrieval: one between 14.5 h and 15 h and the second between 15.4 h and 16 h. The peak values of the retrieved IWC are near 2000 mg m$^{-3}$ (Fig. 10a,b) whereas those of the model simulations do not exceed 400 mg m$^{-3}$ (Fig. 10c,d). The IWC values of the model simulations strongly depend on the snow fall speed which is a constant prescribed in the model. In the present simulations, its value is 1.5 m

s$^{-1}$. Additional sensitivity experiments made by setting its value to 0.6 m s$^{-1}$ led to IWC peak values near 800 mg m$^{-3}$ (not



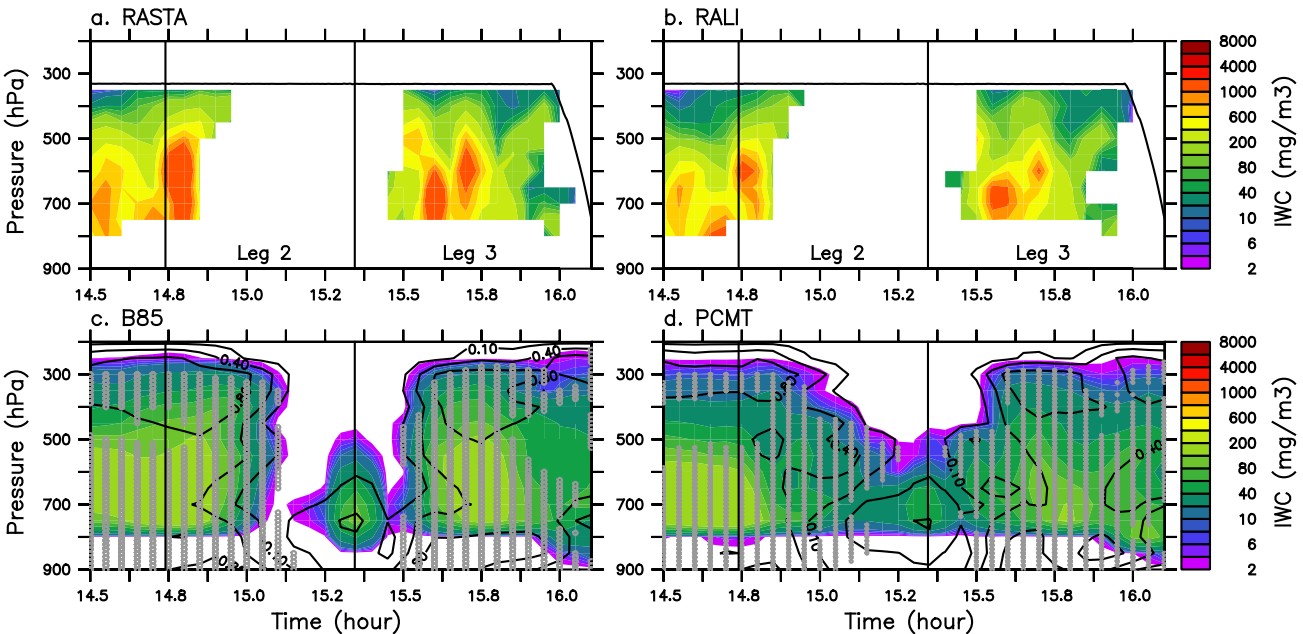

**Figure 10.** Ice water content (mg m$^{-3}$) for a) retrieval from radar RASTA b) retrieval from radar-lidar RALI along legs 2 and 3 of F7 (airborne level in black line); Total ice water content (snow+cloud ice water; mg m$^{-3}$; shading) and cloud fraction (black contours) for c) B85 and d) PCMT respectively. Grey circles represent WCB trajectories crossing F7.

shown). So even in the case of low snow fall velocity, the IWC is largely underestimated in the model. In a supplementary figure (Fig. S5), the IWC divided by the cloud fraction, which could be thought as being more relevant to compare to observations, also fails to reproduce the high IWC values detected in the observations. This result is not surprising following Mazoyer et al. (2021) who found similar underestimation in regional model simulations of the Stalactite cyclone. The comparison between the two simulations shows that the peak values are slightly higher in B85 than in PCMT but the difference is weak to be conclusive.

The difference in IWC spatial distribution is worth commenting, especially with regard to the previous sections. The flight crossed the separating area between the cloudy WCB region with high IWC values and the clear-sky region close to the cyclone center twice, at 14.9 h and 15.5 h. These two transitions are well visible at 15.0 h and 15.5 h in B85 but are much less clear in PCMT. In the latter run, in the observed clear-sky region, there are clouds (Fig. 10d) whose tops are about 500 hPa. Also many trajectories belonging to that area between 15.0 h and 15.5 h satisfy the WCB criterion (see grey circles in Fig. 10c) and correspond to the trajectories which undergo strong heating during their ascent between 900 hPa and 600 hPa (Fig. 6d). Figure S6 shows the same patterns but with model outputs and interpolation made over a 0.1° × 0.1° grid. The results are qualitatively the same but the scales of the clouds are more representative of the model resolution.

To conclude, even though both simulations fail to produce the high values of IWC seen in the observations, the spatial distribution of IWC differs between the two simulations. B85 better represents the abrupt transition between the cloudy region of the WCB and the clear-sky region behind the cold front.



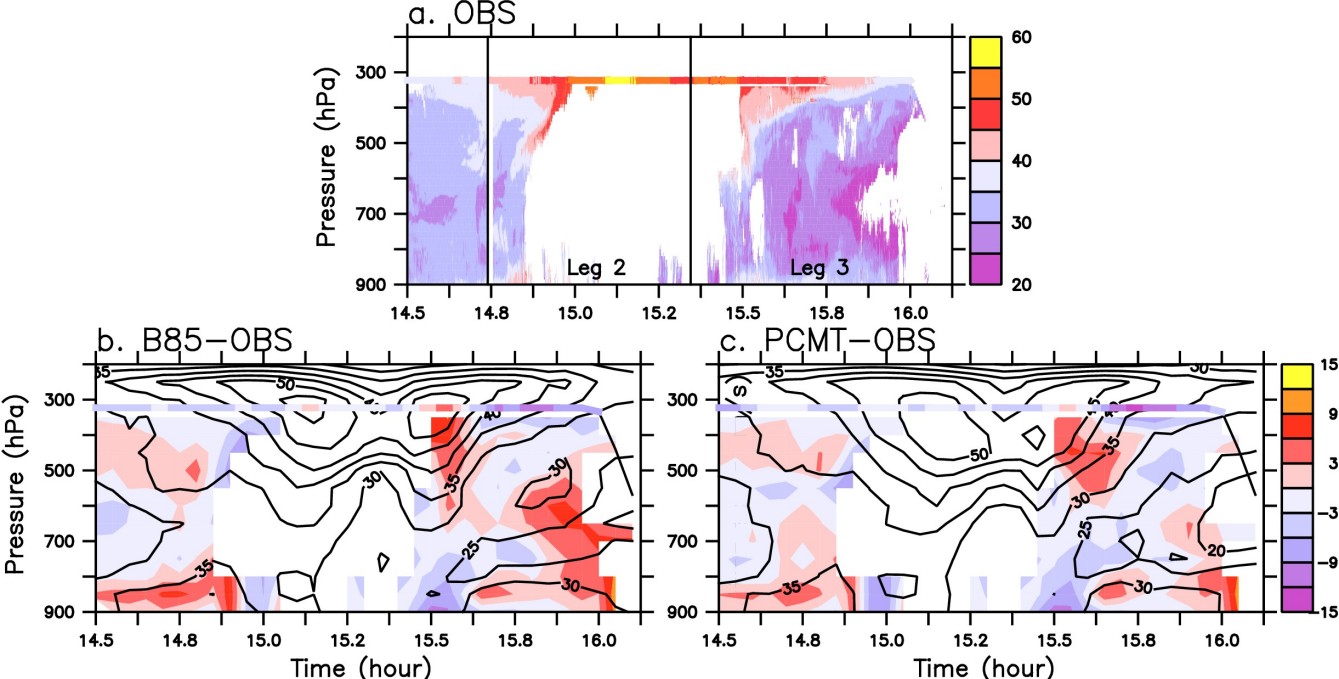

**Figure 11.** (a) Wind speed (shadings) observations (RASTA+ aircraft) at full resolution; Wind speed (black contour) and wind speed anomaly respectively with observations interpolated at model resolution (shadings) for b) B85 (B85-Obs.), c) PCMT (PCMT-Obs.).

To determine which run better represents the jet stream, the wind speed of the simulations is compared with those measured by the radar RASTA and on board the aircraft in Fig. 11. There is generally a good correspondence between the wind speed measured by the aircraft and that measured by the radar (Fig.11a). However, as there are no clouds from 14.9 h and 15.5 h, no

wind speed observations from the radar are available. Hence, the jet stream, which is crossed twice at 15.1 h and 15.5 h, is only very partially covered by the radar measurements (compare the color shadings with the contours). The radar measurements are useless to look precisely at the vertical structure of the jet stream and the aircraft measurements provide information at a given level only. At the aircraft level, the two runs give similar anomalies with some underestimation of winds between 15.6 h and 16.0 h. There is only one region where the two simulations behave very differently: it is between 15.6 h and 16 h and near the

500-700 hPa layer (Figs. 11b-c). While there is a minimum in wind speed in the observations as well as in PCMT in that region, a secondary jet is present in B85 with values near 30 m s$^{-1}$, which was already discussed in section 3. The wind speed in B85 is overestimated by about 6-9 m s$^{-1}$. According to Fig. 5a, this wind speed anomaly is linked to the stronger PV gradient in B85 than PCMT associated with the dipolar PV anomaly located at 600 hPa at the end of the flight (trajectory index from 65 to 75). Comparison with radar measurements leads to the same conclusion as the comparison made with (re)-analyses in section

3. The secondary jet in B85 along leg 3 is not present in any of these references.





## 6  Conclusions

The present study and our companion paper (Rivière et al., 2021, RW21) provide a general view of the impact of deep convection representation in a global numerical weather model on the WCB of an explosive extra-tropical cyclone observed during NAWDEX and on the jet stream aloft. Three simulations of the Météo-France global model ARPEGE, which only differ by
their deep convection representation, are investigated. Two of them use the model with a distinct deep convection scheme activated: one with the scheme developed in Bougeault (1985, B85), the other one with the one from Piriou et al. (2007, PCMT). In the last ARPEGE simulation, called NoConv, no deep convection scheme is activated. The companion paper investigated the general behaviour of WCB activity in the three simulations and its impact on the jet stream in the WCB outflow region above the bent-back warm front. The present study was dedicated to the impact of these parameterization schemes on the jet
stream in the WCB ascending region above the cold front.

The systematic comparison made between the three simulations led to the following conclusions:

- The deep convection representation has an important effect on the vertical structure of the jet stream above the cold front: the jet stream is deeper in NoConv simulation, i.e. without parameterized deep convection, and in PCMT simulation than in B85 simulation.

- The deeper jet stream in NoConv and PCMT compared to B85 is associated with a deepening of the dynamical tropopause (i.e. higher PV) behind the cold front and with more PV destruction ahead of the cold front in middle troposphere (600 hPa). The difference in PV between PCMT and B85 is marked by a dipolar PV anomaly centred on the jet core which reinforces the PV gradient and thus the jet in middle troposphere. The dipolar PV anomaly is due to differences in diabatic processes between the two simulations.

- The same tropopause deepening is observed for PEARP members sharing the same deep convection closure, suggesting, as in RW21, the key role played by that closure on the jet stream structure.

- On the cold-air side of the jet, the high PV area of the dipolar anomaly is due to different behaviours of the Lagrangian trajectories reaching that area. In B85, they form an homogeneous group of trajectories staying at the same level pressure in middle troposphere and undergoing modest PV fluctuations in the cold sector. In contrast, the PCMT trajectories are
clearly separated into two groups. One group of trajectories behave like in B85 with weak pressure variations but are more subject to PV increase because they pass over a more marked cooling due to radiation and turbulence above cold-sector convective clouds. The second group behaves in a totally different manner; they come from the boundary layer, ascent on the western flank of the region of strong latent heating and undergo PV increase, before joining the first group at the same altitude. The strong latent heating is mainly due to large-scale cloud and to a lesser extent to convection.
The trajectories of the second group satisfy the WCB criterion of 300 hPa ascent in 24 h chosen in the present study. The presence of ascending trajectories very near the core of the cold front is unexpected and is associated with a more important overlapping of the heating area and the horizontal temperature gradient in PCMT and NoConv than in B85 (not shown).





- On the warm-air side of the jet, WCB Lagrangian trajectories are quite similar but their behaviour, synthesized in Fig. 12, is clearly different between the two deep convection schemes. With PCMT, WCB trajectories pass sooner through the main heating area as it is located at lower altitude than in B85 (Fig. 12a). Hence, a sooner decrease of PV occurs for the trajectories in PCMT and negative PV tendency appears in middle troposphere (Fig. 12b). This is to be contrasted with B85 where the peak values of the heating extend further upward and much less PV destruction occurs at mid-tropospheric levels. This difference in the altitude of the heating maximum explains the difference in PV ahead of the cold front in middle troposphere.

Then, the question of the realism of the different hindcasts has been addressed by comparing them to different (re)-analyses datasets and to NAWDEX airborne observations. It led to the following conclusions:

- The jet stream structure in (re)-analyses datasets as provided by ERA5 and ECMWF / Météo-France operational analyses lies in between the B85 simulation on the one hand and the PCMT and NoConv simulations on the other hand. For instance, the altitude of jet maximum in B85 is located above those in (re)-analyses datasets which themselves are above those in PCMT and NoConv. Another example concerns the peak values of PV near 600 hPa: in descending order there are the PCMT and NoConv values followed by (re)-analyses datasets and finally B85 values.

- The ice water content retrieved from the radar-lidar measurements using the Varcloud algorithm (Delanoë and Hogan, 2008; Cazenave et al., 2019) shows a clear separation between the cloudy region ahead of the cold front and the ice-cloud free region behind it. This separation is well identified in B85 but not in PCMT. Behind the cold front, B85 is more realistic than PCMT which exhibits too much mid-level clouds.

- Since the main jet is largely embedded in clear-sky regions, the Doppler cloud radar observations are useless to determine which run is more realistic. However, analysis of the wind speed anomalies with respect to the observations in cloudy regions ahead of the cold front indicate that B85 creates a secondary jet in middle troposphere which does not appear in the observations nor in PCMT. Therefore, in that particular region, the PCMT simulation performs better and this is due to the sooner PV destruction in PCMT.

Therefore, the present analysis cannot state which hindcasts better represent the observations as the conclusion is strongly dependent on the regions we are looking at. However, it shows that PCMT and B85 have drastically different behaviours with the former being close to the simulation without parameterized convection. The overall picture provided by the present study and the companion paper is the following. In B85 simulation, the heating is more homogeneously distributed ahead of the cold front and along the bent-back warm front, it extends further up leading to stronger PV destruction in the upper troposphere that accelerates the ridge building in the WCB outflow region. In PCMT and NoConv simulations, the heating is more heterogeneous, especially in NoConv, it extends less in the upper troposphere and the loss of PV happens at a lower altitude ahead of the cold front and makes the jet deeper in that region. Finally, note that such a difference between the two deep convection schemes was also found above the cold front of the following extratropical cyclone on 4-5 October 2016 (NAWDEX IOP7; not shown).



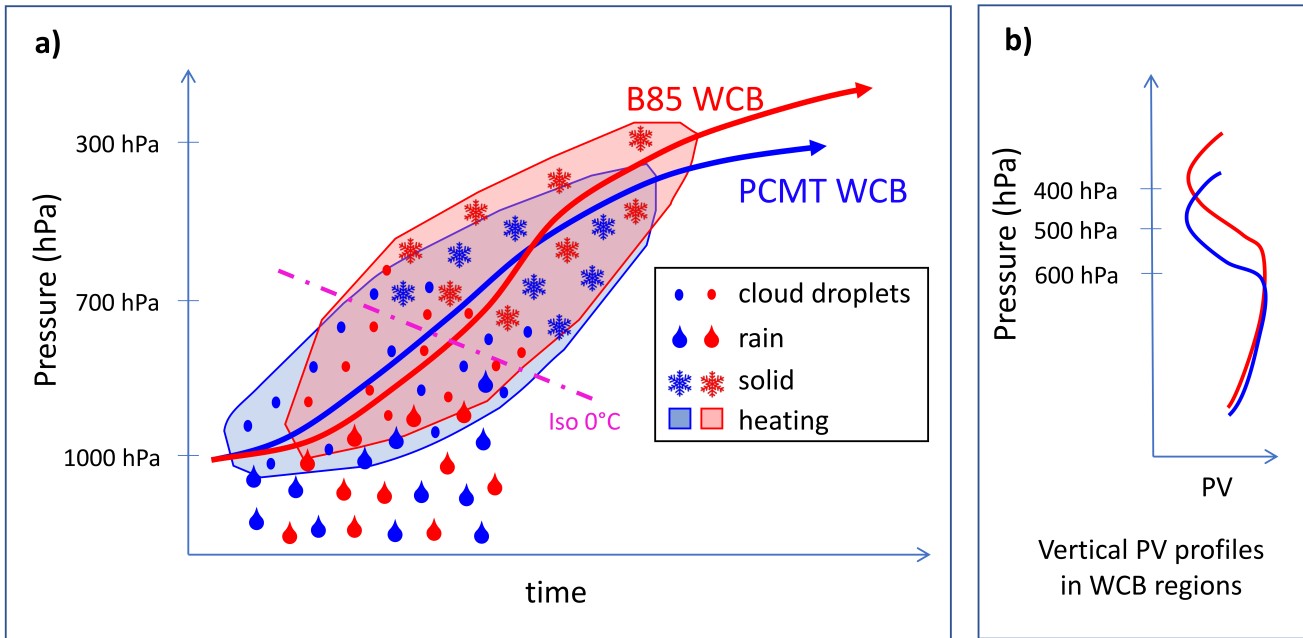

**Figure 12.** Schematic representing differences between the two convection schemes in latent heating and PV tendencies.

The important question that follows up is: where do these different behaviours come from ? A first answer found in this study is the different deep convection parameterization closure used, namely the CAPE for PCMT and moisture convergence closure for B85. However, further sensitivity studies will be planned in order to better identify effects of different parts of the deep convection schemes on mid-latitude cyclogenesis and jet stream.

*Data availability.* Data is available by contacting the corresponding author. ERA5 data are accessible via the climate data store (https://cds.climate.copernicus.eu; DOI: 10.24381/cds.bd0915c6; latest access: April 2021).
Worldview NASA picture is available at https://go.nasa.gov/3xOhRhv

*Author contributions.* GR and PA designed the initial study. MW and GR performed the data analysis and made the figures. MW computed the Lagrangian trajectories. CL performed the ARPEGE simulations with the help of JMP. PA developed the Lagrangian trajectory algorithm. JD, QC, and JP provided the observational datasets. All authors contributed to the scientific discussions.



*Competing interests.* The authors declare that they have no competing interests.

*Acknowledgements.* The study benefited from discussions with the participants of the DIP-NAWDEX (DIabatic Processes in the North Atlantic Waveguide and Downstream impact EXperiment) project which is supported and funded by the Agence Nationale de la Recherche 540 (ANR). It also benefited from discussions with our international NAWDEX partners during annual workshops. ERA5 datasets that have been generated under the framework of the Copernicus Climate Change Service (C3S). We thank Heini Wernli for downloading the ECMWF-IFS operational analysis data and Hanin Binder for sending them to us. We acknowledge the use of imagery from the NASA Worldview application (https://worldview.earthdata.nasa.gov), part of the NASA Earth Observing System Data and Information System (EOSDIS).

*Financial support.* This research has been supported by the Agence Nationale de la Recherche (grant no. ANR-17-CE01-0010-01). The 545 airborne measurements and the SAFIRE Falcon flights received direct funding from IPSL, Météo-France, INSU-LEFE, EUFAR-NEAREX, and ESA (EPATAN, contract no. 4000119015/16/NL/CT/gp).





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
