# Peer review of "Diabatic processes modulating the vertical structure of the jet stream above the cold front of an extratropical cyclone: sensitivity to deep convection schemes"

_Weather and Climate Dynamics, 2021_

## Referee Comment (RC1)

Review of wcd-2021-76

*"Diabatic processes modulating the vertical structure of the jet stream above the cold front of an extratropical cyclone: sensitivity to deep convection schemes"*

by Meryl Wimmer et al.

Paper in review in Weather and Climate Dynamics Discussion

This study presents a detailed analysis of the influence of using two different deep convection parameterization schemes on the wind speed in the mid-troposphere and associated PV structure above the cold front of an extratropical cyclone. Therefore, two simulations with different convection schemes are compared to each other, as well as to three (re-) analysis data sets and airborne observations of ice water content and wind speed. Furthermore, backward trajectories are used to show that differences in the PV structure in both simulations are related to diabatic processes behind and ahead of the cold front. The authors find that using different convection schemes results in differences in the representation of diabatic heating ahead of the cold front, which modifies diabatic PV modification and finally influences the tropopause structure, associated PV gradients, and the jet in the middle troposphere. Although various different datasets are employed in this study, it remains elusive as to which convection scheme is more realistic, as both model simulations are in between the (re-) analyses, both models strongly underestimate ice water content, and both show a bias in the jet structure. While this analysis focuses on one specific time and vertical cross-section only, the (systematic) impact of the different convection schemes is a timely question and fits the scope of Weather and Climate Dynamics. I recommend the publication of this manuscript, however, I have several comments and questions that should be addressed before publication and are listed below.

**1 General Comments**

1 Direct impact of convection schemes for observed differences
While the differences between both simulations with PCMT and B85 in terms of air mass transport and diabatic processes are described in detail, I would appreciate if the differences could be related more closely to both convection schemes throughout the manuscript. For the mid-tropospheric jet, a clear difference in air mass origin is found between PCMT and B85. Could the authors explain in this section how the different heating patterns and thus trajectory pathways are related to the convection schemes? I understand that the companion paper part I deals with this topic more closely and it is briefly mentioned in the introduction, however, I would appreciate a reminder of the actual cause of the differences in the according paragraphs.

2 Focus on one specific location and time

The paper focuses on the differences between the simulations at 15 UTC and at one specific location. I assume this is motivated by the availability of observations at that time and location. However, I was wondering if the observed differences are somewhat representative of the evolving tropopause and jet structure? How do the differences evolve with time? I understand that the trajectories provide a temporal evolution of diabatic processes, but they are still specifically related to the single time step and cross-section. Considering that the simulations are already at a lead time of more than 24 h, how relevant are small differences in timing and spatial shifts between the simulations when the differences for one specific cross-section are evaluated? Do the differences consistently grow with increasing lead time?

3 Mid-tropospheric jet
The major part of this manuscript analyses the difference in wind structure at approximately 600 hPa, and the presence of a secondary jet at that level in some datasets. While the term 'jet' is not necessarily limited to maximum wind speed in the upper troposphere, it is common to define the jet in the upper troposphere (e.g., https://glossary.ametsoc.org/wiki/Jet_stream). While the introduction provides a detailed review of upper-level jet literature, I would appreciate if the authors could also include some introduction about the relevance of these mid-tropospheric jets (e.g., Georgiev and Santurette, 2009, https://doi.org/10.1016/j.atmosres.2008.10.024; Kaplan et al., 2009, https://doi.org/10.1175/2009JHM1106.1). Furthermore, it might be helpful to slightly adjust the title and add that the mid-level jet is one of the foci. This would also more clearly distinguish this manuscript from the companion paper RW21 ('The impact of deep convection representation in a global atmospheric model on the warm conveyor belt and jet stream during NAWDEX IOP6').

4 Length of the paper
The manuscript in its present form is rather long and I think it could be shortened. I would appreciate if in particular parts of sections 3 and 4 could be re-phrased and streamlined to improve readability with a focus on the relevant processes. Please see also specific comments below.

**2 Specific comments**

**ABSTRACT**

1. l. 7: 'jet core in middle troposphere': Based on Fig. 2, I would place the 'jet core' rather between 300 and 400 hPa. How do the authors define 'jet core'?

**1 INTRODUCTION**

2. l. 26 Please define NWP when it is first introduced.

3. l. 40f: 'very short-term forecasts'. Please specify the lead time.

4. l. 41: 'analysed this could affect Rossby wave propagation': This sounds speculative. Please clarify if it was analysed or hypothesized.

5. l. 45ff and l. 53ff: It seems as if the authors split the 'PV tracer' and the 'Lagrangian PV framework' in two separate methods (I may have misunderstood these two paragraphs). If the authors would like to keep these methods separated, please double-check the referenced literature and which of the methods was applied for which study. Some of the mentioned studies apply both methods. Low-level PV anomalies in extratropical cyclones have also been analysed systematically in Attinger et al. (2021; https://doi.org/10.5194/wcd-2-1073-2021).

6. l. 56ff: I appreciate the comprehensive overview of literature here, however, I think re-phrasing this paragraph would help the reader. Instead of writing 'author 1 et al. showed', I would suggest summarizing the existing results in terms of impact on the PV and jet structure.

7. l. 73ff: The description of the results of the companion paper is essential for this study. I suggest using a separate paragraph for the description of the relevant main results from RW21. Please also clearly state the key results relevant for the study at hand.

8. l. 85: 'the heating extends further upward'. Does this relate to total heating from microphysics and convection scheme or convection scheme only?

9. l. 86: 'This leads to a distinct location of the jet stream'. Please clarify 'distinct location'.

**2 DATA and METHOD**

10. l. 144 and 148: 'Figure 1b gives an insight of the cloudy region sampled by the flight.' and 'During this flight, different measurements have been made.' I think these sentences are not necessary, and the manuscript could be shortened by avoiding such sentences. I would kindly ask the authors to streamline these paragraphs.

11. l. 150: 'LATMOS and DT-INSU': Please specify if relevant, else remove.

12. l. 158: The in-situ wind measurements were already mentioned in l. 154.

13. l. 165: Why did the authors decide to use only every 2nd grid point for the ERA5 dataset, instead of also interpolating the data as done for the other datasets?

14. l. 169ff: Why did the authors decide to select trajectories that ascend 300 hPa only? In which vertical levels were the trajectories initialized?

15. l. 175: 'over the whole vertical'. Is there a word missing?

16. l. 175f: 'trajectories are seeded on a vertical regular grid spacing of 12.5 hPa from 975 hPa to 200 hPa and a horizontal grid spacing of about 0.3 in longitude and latitude': Is it meaningful to seed trajectories at a higher vertical and horizontal resolution than the actual data they are computed from (which is 0.5 in the horizontal and 50 hPa in the vertical if I'm correct)?

17. l. 178ff: I appreciate the detail concerning the starting times for the trajectories. Just out of curiosity, did the authors check if the timing is critical for the differences described later? The simulations are already at a lead time of approx. 27 h, and I was wondering how important small shifts in timing are. If trajectories are started a little earlier or later, does PCMT still show two different clusters while B85 only shows one (e.g., Fig. 7)?

18. l. 200ff: Do I understand correctly, the PV tendencies for friction and diabatic heating are computed differently (method 1 for friction vs. method 2 for diabatic heating)? How large are the differences between methods 1 and 2? If they are large, doesn't this introduce uncertainties if the PV tendencies from friction and heating are compared?

**SECTION 3**

19. l. 213ff: Interestingly, ERA5 shows more negative PV in the cross-section shown at 15 UTC, while in the companion paper RW21 Fig. 8 suggests that the ARPEGE analysis is characterized by enhanced occurrence of negative PV at 12 UTC. Are these differences related to specific timing, location, or selected vertical level?

20. l. 233: Please mention which datasets.

21. l. 256: Is it correct that member 3 uses B85 but with a CAPE closure? Please clarify in the manuscript.

22. l. 246ff: I find the results in this section very interesting, because it provides a more systematic assessment and does not only show two different simulations. In the supporting material, it is difficult to compare all of the members. Did the authors try to compute vertical cross-section composites for the members in both clusters (i.e., moisture convergence vs CAPE closure)? I understand that averaging may smooth out some differences, but I would be interested if systematic differences remain between both clusters.

23. l. 261: 'smaller negative PV values to the east in the mid-troposphere around 600 hPa': I cannot see a systematic difference of neg. PV between the datasets in Fig. 3. It appears very patchy and non-systematic in the simulations and the (re-)analyses. Is this a relevant difference?

**SECTION 4**

24. l. 273ff: What is the relevance of the curvature of the trajectories, in particular, if there is no clear separation? As this is not used in the following, I would suggest removing this paragraph.

25. l. 282ff: Please streamline this short paragraph.

26. l. 299: Why was neither time nor longitude used for the abscissa? I'm not sure, I fully understand what exactly 'number of trajectory seeds' refers to? Is this a pseudo-longitude?

27. l. 323: It is clear from Fig. 5, but I think it would be helpful if the authors mentioned that the jet between 500 and 700 hPa is analysed.

**SECTION 4.1**

28. l. 325ff I would suggest to restructure this section and to discuss the origin of trajectories first, instead of directly starting with PV evolution and detailed tendencies. While reading, I immediately wondered if the large difference in PV results from different ascent pathways, which is discussed in the manuscript at a later stage. I would find it more intuitive and easier to follow if switched around. This way, it would also be more straightforward to understand the PV evolution.

29. Fig. 6: I assume Fig. 6 without considering the ascending trajectories in PCMT would result in a similarity between B85 and PCMT? Is this correct, i.e., are the major differences only caused by the ascending portion of trajectories?

30. l. 336ff: How close do the individual PV tendencies get to closing the budget, i.e., if the tendencies are summarized, how large is the deviation between a) and b)?

31. l. 342ff: Although a little smaller than the heating induced PV tendency, I think the frictional PV tendency is still relevant for the PV budget in PCMT. Is the difference between the frictional PV tendencies caused by the ascending trajectories in PCMT (Figs. S3 and S4 suggest that at least turbulent heating is most important below 600 hPa)?

32. l. 344ff: Considering that all trajectories in B85 and a large number of trajectories in PCMT remain at the same pressure level and do not ascend, I find it critical to define these as WCB trajectories. After these trajectories crossed the flight track, do they continue to ascend? I would suggest to re-name this cluster of trajectories or thoroughly discussing this topic.

33. l. 347f: The PV framework and diabatic PV modification have not been explicitly introduced in the introduction. I think it is also not necessary in this manuscript, in particular, because it is introduced in RW21. However, I think some references and one explanatory statement may be useful here for readers who are less familiar with this concept.

34. l. 359f: 'the two PV start to get away from each other': Please re-phrase.

35. l. 364: I find it interesting that the convection schemes impact substantially the turbulence scheme and the structure of large-scale cloud heating (Figs. S3 and S4). Do the authors understand why and how?

36. l. 372ff: I appreciate the summary paragraphs at the end of each subsection. Could the authors discuss additionally how the observed differences are related to the different convection parameterization schemes? It summarizes the difference in terms of large-scale heating and PV modification, however, it lacks to discuss the underlying causes (i.e., different convection parameterization).

**SECTION 4.2**

37. l. 382: I assume the backward trajectories considered in the following are the green dots in Fig. 5. If this is correct, it may be helpful to mention here.

38. l. 389f: I agree, but I think that the frictional PV tendency for PCMT is non-negligible as it strongly reduces the differences between PCMT and B85.

39. l. 402f: The authors describe an interesting systematic tilt of the heating in PCMT. Do the authors understand or have a hypothesis what causes these differences in heating pattern between PCMT and B85?

40. l. 410: 'heating which is more confined at lower levels': this could be an option to link the general WCB analysis (section 4) to the specific analyses in sections (4.1 and 4.2) through refering back to Fig. 4 and dicussing the differences in heating pattern.

**SECTION 5**

41. The large IWC differences in magnitude between simulation and observations are interesting. Were such large differences expected by the authors? Do the authors think that this is disconcerting? How does IWC from the (re-)analyses compare with (i) the simulations, and (ii) the observations? How reliable are the retrievals? Apart from Mazoyer et al. (2021), have other studies compared the estimated IWC with model results?

**CONCLUSIONS**

42. l. 469f: I suggest to add 'mid-level' jet or similar for clarification for the reader and to distinguish this manuscript from the companion paper RW21.

43. l. 490: Here only the second ascending group of trajectories is referred to as WCB trajectories, whereas above, all trajectories are refered to as WCB. Please be consistent and adjust throughout the manscuript. See also comment to l. 344.

44. l. 491ff: I find this discussion very interesting and would appreciate if theses differences between heating in B85 and PCMT are more thoroughly also discussed throughout section 4 in the appropriate places.

45. l. 494ff: The differences between trajectories and PV modfication are clearly summarized here. I would appreciate if section 4 could be rephrased and streamlined to highlight the important differences.

46. I appreciate the comprehensive summary in the conclusions and the schematic in Fig. 12.

**FIGURES**

47. Fig. 1: It would be nice to additionally show the location of the jet (e.g., wind or selected PV contours) because the manuscript focus on the jet and PV structure. Is it correct that Fig. 1b does not show the same region as Fig. 1a? I would appreciate if Fig. 1b shows at least the approximate longitude and latitude coordinates. I was additionally wondering why 12

UTC is shown in this figure although the following figures all focus on 15 UTC.

48. Fig. 2: right bracket is missing in caption after 'in Fig. 1'

49. Fig. 3: Please add that the $0\,\mathrm{PVU}$ contour is shown by bold contour (if this is correct?).

50. Fig. 5: How do the authors define positive and negative 'anomalies' along trajectories? I would find it helpful if in a) and b) at least one PV contour of PCMT and B85 would be included in both panels. This would help the interpretation of the differences shown in shading.

51. Fig. 7a,b: Is the heating vertically averaged? Please clarify in the caption. Note also the typo in $K\,h^{-1}$. Please also note comment to l. 344 about WCB trajectories. Please re-consider if the quasi-isentropic trajectories should be refered to as WCB trajectories.

52. Fig. 11: Please clarify the caption.

**3   Technical corrections**

1. general comment: I would suggest to avoid rather technical terms in the manuscript, such as 'plot', 'vertical point of view', '3D picture'.

2. Please try to be consistent with the wording. I think starting from l. 301 the authors refer to the differences between the simulations as 'anomalies'.

3. l. 37: I'd suggest to rephrase to either 'NAWDEX's objective' or the 'objective of NAWDEX'

4. l. 92: 'number of respects'. I think you mean 'number of aspects'

5. l. 115: 'an horizontal resolution': Change to 'a horizontal resolution'. See also l. 154.

6. l. 141: 'Figures 1a and b show the position of the flight according to the Stalactite cyclone': Please re-phrase this sentence, e.g., 'Figure 1 shows the position of the flight in relation to the cyclone'

7. l. 269: I would replace 'modelized' by 'modeled'.

8. Table 1: Typo in unit for wind speed.

9. Fig. 3: I'd suggest to change 'without any deep convection representation' to 'explicit deep convection'.

---

## Author Comment (AC1)

**Manuscript WCD-2021-76**

**5**

**cold front of an extratropical cyclone: sensitivity to deep convection schemes" Authors' response**

**Dear Editor,**

We would like to thank first the two referees for their deep analyse and their relevant remarks that helped to improve the quality
of our manuscript. Please find below our point by point answer to the reviewers' comment. Replies to reviewers are in blue, while reviewers' comments are in black.

"Diabatic processes modulating the vertical structure of the jet stream above the

**Reply to referee 1**

- 15 This study presents a detailed analysis of the influence of using two different deep convection parameterization schemes on the wind speed in the mid-troposphere and associated PV structure above the cold front of an extratropical cyclone. Therefore, two simulations with different convection schemes are compared to each other, as well as to three (re-) analysis data sets and airborne observations of ice water content and wind speed. Furthermore, backward trajectories are used to show that differences in the PV structure in both simulations are related to diabatic processes behind and ahead of the cold front. The
- 20 authors find that using different convection schemes results in differences in the representation of diabatic heating ahead of the cold front, which modifies diabatic PV modification and finally influences the tropopause structure, associated PV gradients, and the jet in the middle troposphere. Although various different datasets are employed in this study, it remains elusive as to which convection scheme is more realistic, as both model simulations are in between the (re-) analyses, both models strongly underestimate ice water content, and both show a bias in the jet structure. While this analysis focuses on one specific time and
- 25 vertical cross-section only, the (systematic) impact of the different convection schemes is a timely question and fits the scope of Weather and Climate Dynamics. I recommend the publication of this manuscript, however, I have several comments and questions that should be addressed before publication and are listed below.

We would like to thank the referee for accepting to review our second paper and for his comments that helped to improve the content and clarity of the paper. The point by point answers to the comments are hereafter provided.

**General Comments**

45

55

- 1. Direct impact of convection schemes for observed differences
- While the differences between both simulations with PCMT and B85 in terms of air mass transport and diabatic processes are described in detail, I would appreciate if the differences could be related more closely to both convection schemes throughout the manuscript. For the mid-tropospheric jet, a clear difference in air mass origin is found between PCMT and B85. Could the authors explain in this section how the different heating patterns and thus trajectory pathways are related to the convection schemes? I understand that the companion paper part I deals with this topic more closely and it is briefly mentioned in the introduction, however, I would appreciate a reminder of the actual cause of the differences in the according paragraphs.

The origin of the differences in heating and trajectory pathways are closely linked to the deep convection closure as shown in the supplementary figures. All the simulations run with the CAPE closure (e.g., PCMT) resemble the simulation where no deep convection is activated and behave differently than the simulations run with the moisture closure (e.g., B85). Our conclusion is that the CAPE is not strong enough along such warm conveyor belts (WCBs) to trigger the convection scheme while the moisture convergence contains a synoptic-scale pattern that triggers the convection scheme more systematically in the WCBs. These results were already found when looking in the WCB outflow region in the first paper and are recalled in the introduction. In the present paper, we show similar behavior above the cold front in the ascending branch of the WCB which has some impact on the representation of the vertical structure of the jet.

- 50 In future studies, sensitivity experiments will be performed to more systematically investigate the sensitivity to the closure and get a deeper insight of its effect along WCBs and the jet stream.
  - 2. Focus on one specific location and time

The paper focuses on the differences between the simulations at 15 UTC and at one specific location. I assume this is motivated by the availability of observations at that time and location. However, I was wondering if the observed differences are somewhat representative of the evolving tropopause and jet structure? How do the differences evolve with time? I understand that the trajectories provide a temporal evolution of diabatic processes, but they are still specifically related to the single time step and cross-section. Considering that the simulations are already at a lead time of more than 24 h, how relevant are small differences in timing and spatial shifts between the simulations when the differences for one specific cross- section are evaluated? Do the differences consistently grow with increasing lead time?

Even though the differences between simulations are shown for a given time and a specific location along the cold front they are representative of systematic differences. Indeed, similar differences appear when looking at different lead time and for other hindcast simulations starting at different initial times. Figure 1 (below) shows the wind speed at 600 hPa for B85 (first column) and PCMT (second column) and also the difference of the PV between the two runs (third column) by initializing the simulations at three distinct initial times: 00 UTC 1 Oct, 12 UTC 1 Oct and 00 UTC 2 Oct. In each case, the wind speed is stronger in PCMT than B85 and is associated with a band of positive PV difference to the west and a band of negative PV difference to the east along the cold front. For the run starting earlier (00 UTC 1 Oct), a third band is visible with positive values of the PV difference. According to figure 1, differences seem to increase with lead time. Interestingly, hindcast simulations of another cyclone (see figure 2 below) show the same difference between PCMT and B85 and a clear tripolar PV anomaly is visible (Fig. 2c). Therefore these differences seem to be systematic and are not specific to the time and location presented in the paper. In the revised paper, we mention these different hindcast simulations to underline the recurrence of these differences.

**3. Mid-tropospheric jet**

75

80

85

95

The major part of this manuscript analyses the difference in wind structure at approximately 600 hPa, and the presence of a secondary jet at that level in some datasets. While the term 'jet' is not necessarily limited to maximum wind speed in the upper troposphere, it is common to define the jet in the upper troposphere (e.g., https://glossary.ametsoc.org/wiki/Jet stream). While the introduction provides a detailed review of upper-level jet literature, I would appreciate if the authors could also include some introduction about the relevance of these mid-tropospheric jets (e.g., Georgiev and Santurette, 2009, https://doi.org/10.1016/j.atmosres.2008.10.024; Kaplan et al., 2009, https://doi.org/10.1175/2009JHM1106.1). Furthermore, it might be helpful to slightly adjust the title and add that the mid-level jet is one of the foci. This would also more clearly distinguish this manuscript from the companion paper RW21 ('The impact of deep convection representation in a global atmospheric model on the warm conveyor belt and jet stream during NAWDEX IOP6').

We would like to thank the referee for mentioning these two references that we were not aware of. However, the main band of wind speed maximum discussed in the present paper (see e.g., Fig.1 of the present document or Fig. 3 of the paper) corresponds to the lower part of the upper-level jet and not to a mid-tropospheric jet per say. It is only the secondary band of wind maximum to the northeast of the main one that appears to be a mid-tropospheric jet in some simulations (e.g., see Fig. 5a of the paper near the trajectory index 70). But this secondary jet is not the focus of the present study. For these reasons, we think it is better not to go into the details of mid-tropospheric jet literature in the introduction and we decided not to change the title of the manuscript. In the revised paper, the two references will be cited when the secondary jet is described but no deep investigation on that jet is provided.

90 4. Length of the paper

The manuscript in its present form is rather long and I think it could be shortened. I would appreciate if in particular parts of sections 3 and 4 could be re-phrased and streamlined to improve readability with a focus on the relevant processes. Please see also specific comments below.

As suggested in the following specific comments, some paragraphs have been shortened and some sentences deleted. In particular, the text describing Figs. 2 and 3 has been reduced and provides less details to more directly go the point.

**Specific comments**

**ABSTRACT**

- 1. 1. 7: 'jet core in middle troposphere': Based on Fig. 2, I would place the 'jet core' rather between 300 and 400 hPa. How do the authors define 'jet core' ?
- 100 By 'jet core', we mean area with high values of wind speed. However that formulation brings confusing information and thus has been removed.

---

## Author Response (AR2)

**Manuscript WCD-2021-76**

**"Diabatic processes modulating the vertical structure of the jet stream above the cold front of an extratropical cyclone: sensitivity to deep convection schemes"**

**Authors' response**

Dear Editor,
We would like to thank the referee for these new technical corrections. Please, find below our point by point answer to the reviewer's comment. Replies are in blue, while reviewer's comments are in black. To make our answer clearer, we refer to lines in the new revised version of the manuscript.

**Reply to referee 1**

I would like to thank the authors for the careful consideration of the open questions and feedback by the reviewers and for the additional analyses and figures shown in the reply document. The additional figures shown in the reply document and the according modifications in the manuscript strengthen the results and conclusions of the original manuscript. Based on the authors' replies and the revised manuscript, I recommend publication of the manuscript in Weather and Climate Dynamics.

We would like to thank the referee for this second review and these last technical corrections, that have been taken into account in the new submitted manuscript. Our point by points answer to your comments is listed below.

Some technical corrections that could be considered prior to publication:

1. l. 232 Please re-phrase: "but is upper in the two analyses"

   We change the formulation "but is upper" by "with higher heights". Thus the sentence at line 215 in the new submitted manuscript is now : "The height of maximum wind speed fluctuates from dataset to dataset with higher heights in the two analyses than in ERA5".

2. l. 450 Please correct "vertically heating difference" (e.g., change to "differences in vertical heating")

   As suggested, we change the formulation to "difference in vertical heating". Thus, the sentence at lines 411-412 is now : "Such difference in vertical heating has already been observed in Fig. 4 and is partly linked to a different behaviour of deep convection schemes in the liquid phase".

3. l. 452 Please use either singular or plural: "a negative PV differences"

   We correct the sentence and use singular : "a negative PV difference". You will find the corrected formulation at line 413.

4. In the conclusion, "sooner" was replaced by "earlier". I suggest to adjust this throughout the manuscript (see p. 2, 10 and 14)

   To be more consistent, we replace all "sooner" by "earlier" at lines 286, 411 and 496.